# Overtone photothermal microscopy for high-resolution and high-sensitivity vibrational imaging

Le Wang[1,4], Haonan Lin [1,4], Yifan Zhu[2], Xiaowei Ge [1], Mingsheng Li[1], Jianing Liu[1], Fukai Chen[3], Meng Zhang[1] & Ji-Xin Cheng [1,2,3] ✉

Photothermal microscopy is a highly sensitive pump-probe method for mapping nanostructures and molecules through the detection of local thermal gradients. While visible photothermal microscopy and mid-infrared photothermal microscopy techniques have been developed, they possess inherent limitations. These techniques either lack chemical specificity or encounter significant light attenuation caused by water absorption. Here, we present an overtone photothermal (OPT) microscopy technique that offers high chemical specificity, detection sensitivity, and spatial resolution by employing a visible probe for local heat detection in the C-H overtone region. We demonstrate its capability for high-fidelity chemical imaging of polymer nanostructures, depth-resolved intracellular chemical mapping of cancer cells, and imaging of multicellular *C. elegans* organisms and highly scattering brain tissues. By bridging the gap between visible and mid-infrared photothermal microscopy, OPT establishes a new modality for high-resolution and high-sensitivity chemical imaging. This advancement complements large-scale shortwave infrared imaging approaches, facilitating multiscale structural and chemical investigations of materials and biological metabolism.

The shortwave infrared (SWIR) window, typically spanning from 900 nm to 2 μm in the electromagnetic spectrum, offers distinct advantages for bioimaging. Compared to the mid-infrared window, the SWIR window has an ~20–5000 times (wavelength-dependent) smaller water absorption coefficient[1]. Furthermore, SWIR imaging allows for unprecedentedly large penetration depth attributed to much-reduced light scattering[2,3]. This depth is wavelength-dependent and can extend to the millimeter level[4,5]. With the rich chemical information based on overtone absorption, SWIR spectroscopy emerges as an appealing analytical tool[6]. Several SWIR imaging methods have been developed, including hyperspectral reflectance/transmittance imaging[7–9], diffuse optical spectroscopic imaging (DOSI)[10–12], and photoacoustic microscopy (PAM)[13,14]. SWIR hyperspectral imaging, which measures reflectance or transmittance, primarily serves in the qualitative spectral

characterization of samples at the macroscale, focusing on properties related to absorption and scattering. To probe macroscopic sample areas for improved representative sampling, this technique generally sacrifices spatial resolution in favor of a larger field-of-view (FOV) due to the space-bandwidth product limit, thereby leading to resolutions on the order of ten micrometers. Diffraction-limited spectroscopic imaging can be attained by magnifying the FOV onto the camera or implementing a laser point-scanning design[15,16]. DOSI, adopting a wide-field approach, measures the combined effects of broadband optical absorption and scattering. It has proven successful in the study of thick tissues, such as functional information of bone sarcomas and breast tumor hemodynamic responses. This method offers a sub-millimeter lateral resolution and a penetration depth of generally a few millimeters. PAM utilizes an objective lens as the focusing element and

[1]Department of Electrical and Computer Engineering, Boston University, Boston, MA 02215, USA. [2]Department of Chemistry, Boston University, Boston, MA 02215, USA. [3]Department of Biology, Boston University, Boston, MA 02215, USA. [4]These authors contributed equally: Le Wang, Haonan Lin. ✉e-mail: jxcheng@bu.edu

collects acoustic waves with an ultrasonic transducer to achieve optical resolution. SWIR PAM was first developed for imaging lipids based on the 2nd overtone of the C-H bond[14] and more recently imaging water at 1930 nm[17], whereas the sensitivity is heavily reliant on the quality of the ultrasonic transducer employed[18].

Here, we introduce a technique named overtone photothermal (OPT) microscopy, which excites the sample in the SWIR window and employs a visible probe to detect the thermal lensing effect caused by overtone vibrational absorption. Specifically, we utilized a femtosecond laser tunable in the C-H 2nd overtone window and a 520 nm probe beam through the second harmonic generation of a 1040 nm femtosecond laser. Both pulses were chirped to picosecond to minimize photodamage. Compared to existing SWIR imaging methods, OPT microscopy provides simultaneous high resolution and high sensitivity. Spectroscopic OPT imaging was successfully demonstrated on polymer nanostructures. In conjunction with spectral unmixing algorithms, depth-resolved OPT mapping of protein and fatty acids in cancer cells, *C. elegans*, and brain tissues was performed at intracellular and multicellular levels. These results underscore the potential of OPT microscopy as a valuable technique for investigating chemical structures at high resolution in biological systems and materials.

## Results

### OPT microscope setup and signal extraction

Figure 1a illustrates a lab-built OPT microscope. The specific optical components and signal extraction methods are detailed in the Methods Section. In brief, the OPT microscope utilizes a pulsed SWIR pump beam, tunable in the range of 1080 nm to 1280 nm, and a 520 nm probe beam generated from the second harmonic generation of the 1040 nm output. Both pump and probe beams are chirped to picosecond duration by glass rods. The beams are treated as pseudo-continuous waves with a repetition rate of 80 MHz, and the excitation

beam is modulated at desirable frequency (hundreds of kHz to 1 MHz) and duty cycle (10% to 50%) based on the sample's thermal decay properties. The two beams are delivered collinearly to a laser-scanning upright microscope and focused onto the sample plane using a water immersion objective with a numerical aperture (NA) of 1.2.

Upon the SWIR illumination, the sample's selective absorption within the focal volume leads to a local temperature rise and a thermal gradient, inducing a subtle decrease in the local refractive index at the pump beam focus. To efficiently generate photothermal signals, an axial focus displacement between the two beams is implemented, as depicted in Fig. 1b[19]. In this scenario, the induced thermal lens modifies the propagation of the 520 nm probe beam, causing it to diverge or converge depending on its focal position relative to the pump focus. When there is no axial offset, the thermal lens forms precisely at the focal position of the probe beam. Therefore, the ray locus of the probe beam will not be modified, and the photothermal signal vanishes.

By incorporating an iris with an adjustable NA on the collection condenser, the intensity modulation of the probe beam, corresponding to the selective absorption, can be read out by a photodiode and demodulated using a lock-in amplifier. As shown in Fig. 1c, the green curve represents the transient probe beam intensity change as a function of periodic heating and cooling processes. The direction of the probe intensity change is determined by the direction of the vertical focus displacement of the excitation and probe beams. For the provided thermodynamic trace of dimethyl sulfoxide (DMSO) shown in Fig. 1c, the focus of the probe beam is positioned above that of the excitation beam, resulting in an equivalently divergent lens and a decreased probe intensity on the photodiode. The OPT signal at the modulation frequency is extracted by a lock-in amplifier (Fig. 1d). Spectroscopic OPT imaging is performed by tuning the SWIR wavelength and recording OPT images through laser scanning facilitated by 2D galvo mirrors.

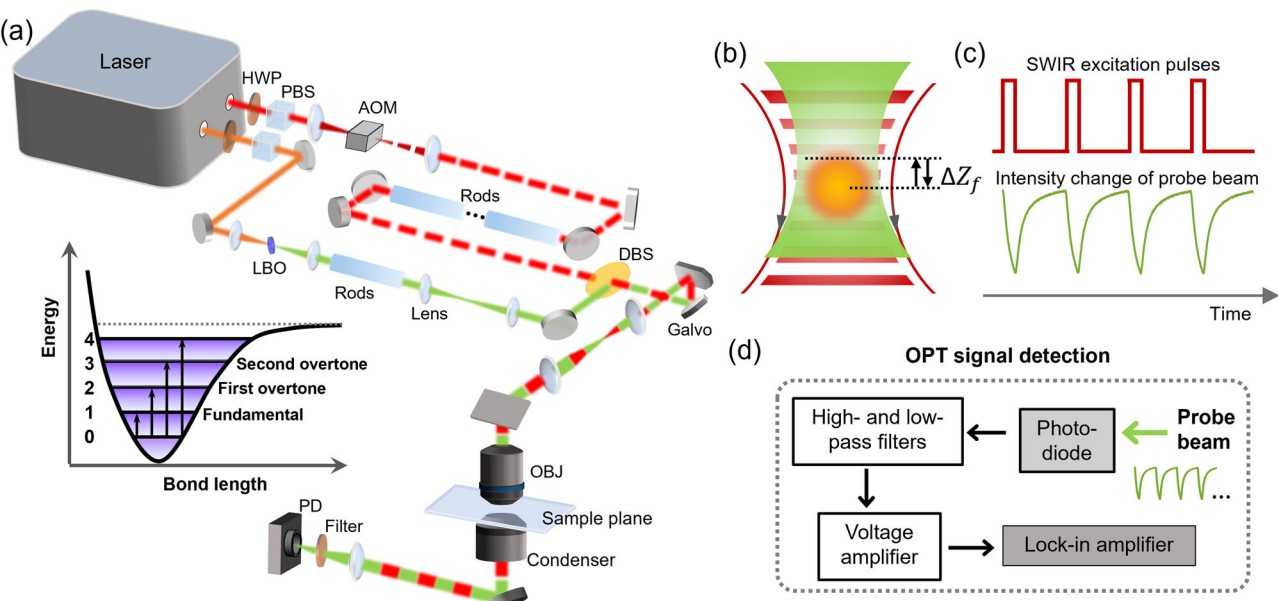

**Fig. 1 | Operational scheme of overtone photothermal (OPT) microscopy.**
**a** Schematics and optical components of the setup. HWP half-wave plate, PBS polarization beam splitter, AOM acousto-optic modulator, LBO lithium triborate crystal, DBS dichroic beam splitter, OBJ objective, PD photodiode. The inset is a vibrational potential energy diagram. **b** Beam propagation in OPT microscopy. The pulsed SWIR excitation beam and the continuous 520 nm probe beam are collinearly propagated and tightly focused onto the sample plane. The axial foci of the two beams have an offset of $\Delta Z_f$. **c** The waveforms of probe intensity as a function

of periodic heating and cooling processes. The sign of the probe intensity change is determined by the sign of the axial offset between the excitation and probe foci. The shown thermodynamic trace was measured on DMSO. **d** Signal extraction in OPT microscopy. The probe intensity change was captured by a photodiode and converted into an electronic voltage signal. After passing through electronic filters and amplifiers, the periodic signal was demodulated by a lock-in amplifier and then registered as OPT signals per imaging pixel.

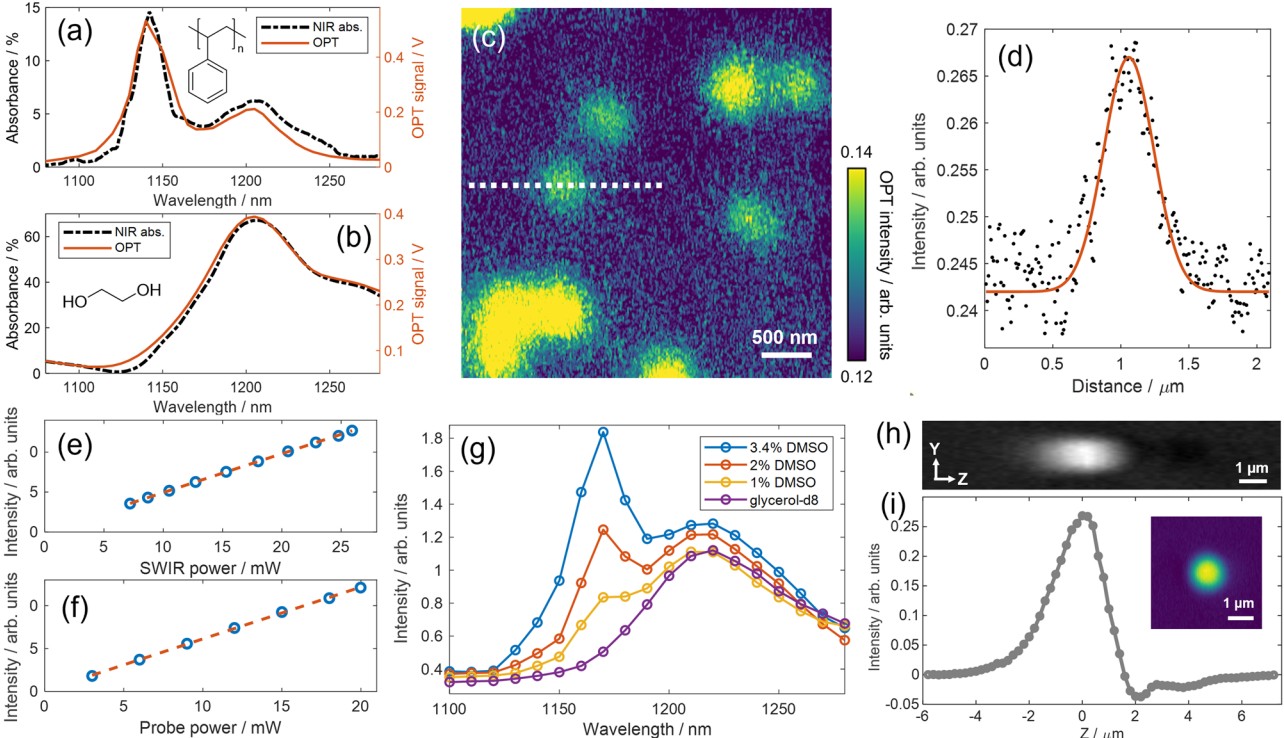

**Fig. 2 | System characteristics. a** Spectral fidelity demonstrated on 10-μm polystyrene beads. The NIR spectrum of PS solution collected from a UV-Vis-NIR spectrophotometer was plotted as a reference. **b** Spectral fidelity demonstrated on pure solvent ethylene glycol. **c** OPT image at 1170 nm of 200-nm PMMA beads dispersed in glycerol-d8. Individual beads and bead aggregates are observed in the image. **d** Cross-section profile along the marked dashed line in **c** to estimate the lateral resolution in OPT microscopy. **e** Power dependence of OPT signals as a function of SWIR excitation power. **f** Power dependence of OPT signals as a function of the 520 nm probe power. **g** OPT spectra measured from multiple concentrations of DMSO@glycerol-d8 solution. The OPT signal at 1170 nm of 1% DMSO can be clearly distinguished out of the background of glycerol-d8. **h** Longitudinal view of a 3D OPT imaging of a 1-μm PMMA bead at 1170 nm. **i** Axial OPT point spread function profile measured from **h**. The inset shows an image of this bead at $Z = 0$ μm. The axial position of the particle was precisely controlled through a piezo stage.

## System characteristics

To evaluate the performance of OPT microscopy, we first conducted a study examining its spectral fidelity. OPT spectra of 10-μm polystyrene (PS) beads and pure ethylene glycol were collected and compared with standard near-infrared absorption spectra from a UV-Vis-NIR spectrophotometer. Our results, as presented in Fig. 2a, b, demonstrate a good consistency between the two types of spectra. The PS spectra exhibited two prominent peaks at 1144 nm and 1205 nm, corresponding to the aromatic and aliphatic C-H 2nd overtone absorption, respectively[4]. The OPT spectra of ethylene glycol showed a broad peak at around 1205 nm, attributed to the peak shape alternation of aliphatic C-H bonds connected with hydroxyl groups.

Additionally, we assessed the lateral resolution and detection sensitivity via OPT imaging of 200-nm PMMA beads. The OPT image was acquired at 1170 nm, corresponding to the 2nd overtone resonance of PMMA. The heating beam had a modulation frequency of 800 kHz and a duty cycle of 50%, resulting in a pulse duration of 625 ns. The average power of the 1170 nm beam was set at 40 mW on the sample. It is important to note that the selection of experimental parameters plays a crucial role in optimizing imaging performance. To guide the parameter selection, we performed a theoretical simulation using COMSOL to analyze the transient temperature changes during the periodic heat accumulation and dissipation processes[20–22]. A detailed description of the thermodynamic simulation can be found in Supplementary Note S1, and thermodynamic traces of 200-nm PMMA beads in glycerol-d8 and D₂O were compared. The simulation revealed that the maximum temperature rise in glycerol-d8 as the medium was ~3.6 K, which is higher than that in

D₂O (Supplementary Fig. S1). The thermal decay constant was found to be 35 ns in glycerol-d8 and 28 ns in D₂O. We found that a repetition rate as high as 1 MHz can be used to enhance the imaging speed, with the prerequisite of ensuring sufficient heat dissipation.

To determine the lateral resolution, we imaged individual PMMA beads and bead clusters at 1170 nm (Fig. 2c). A BM3D denoising algorithm was applied to enhance the signal-to-noise ratio (SNR)[23], and a SNR of 4 was observed on a single bead. The cross-section line profile marked by the dashed line shows a full-width-at-half-maximum (FWHM) of 432 nm (Fig. 2d) and indicates a lateral resolution of 405 nm after deconvolution with the particle size. The theoretical diffraction limit calculated based on two axially overlapped Gaussian beams is given by

$$Resolution = \frac{0.61}{NA} \times \left( \sqrt{1 / \left( \frac{1}{\lambda_{pump}^2} + \frac{1}{\lambda_{probe}^2} \right)} \right) = \frac{0.61}{1.2} \times \left( 1 / \sqrt{\frac{1}{1170^2} + \frac{1}{520^2}} \right) = 242 nm.$$

This discrepancy between theory and experiment is likely due to the practical axial displacement $\Delta Z_f$ between the two beams, and $\Delta Z_f$ can have both positive and negative values, typically on the order of several hundreds of nanometers to several microns[19,24].

The power dependence of OPT signal intensity on the pump and probe power is depicted in Fig. 2e, f. The OPT intensity exhibited a linear relationship with both excitation and probe powers, consistent with observations made in visible photothermal and mid-infrared photothermal (MIP) microscopies[25,26]. To determine the limit of detection, we conducted measurements on serial-dilution samples of DMSO dissolved in glycerol-d8. A peak at 1170 nm, corresponding to the absorption of methyl groups in DMSO, was observed (Fig. 2g).

The OPT spectrum of DMSO was shown in Supplementary Fig. S2a. It should be noted that the solvent glycerol-d8 is not entirely transparent in this spectral window and exhibits a broad peak at approximately 1220 nm, resulting from the resonance of C-D bonds affected by O-D groups. Despite the solvent background, we were able to detect 1% DMSO at a SNR of 8. The theoretical detection limit of DMSO-in-glycerol-d8 solutions, calculated from the statistical 3-Sigma (three times of the measurement noise after background subtraction) at 1170 nm, is found to be 0.3% DMSO (Supplementary Fig. S2b, c). The 200-nm PMMA beads presented in Fig. 2c also highlight the high sensitivity of OPT for imaging of nanoparticles.

To demonstrate the 3D imaging capability of OPT microscopy, we performed depth-resolved imaging of a single 1-μm PMMA bead at 1170 nm, as shown in Fig. 2h. Each image in the stack was collected with a 200 nm increment in the vertical direction. An intensity profile along the axial direction (Fig. 2i) indicates an axial resolution of 2 μm. The dispersive axial profile of OPT signals arises from the interference between scattered and incident laser fields and is closely linked to the axial displacement $\Delta Z_f$ between excitation and probe lasers[27].

## OPT imaging and spectroscopy of fabricated polymer structures

The high spatial resolution and sensitivity of OPT microscopy allow for the identification of fabricated polymer structures. Phase separation patterns of spin-coated PS-PMMA polymer blends (Fig. 3a) and fabricated PMMA striped structures were used to evaluate its performance. Controlled phase separation morphology in polymers is essential for designing thin film devices with desired mechanical, thermal, and electronic properties[28,29]. The resulting morphology can be finely tuned by manipulating factors such as temperature, molar ratio of polymer components, polymer solubility, and film thicknesses[30].

Figure 3b displays OPT spectra of the two polymer components, and Fig. 3c–f presents two-color imaging results of the phase separation patterns. PS domains at 1140 nm and PMMA domains at 1170 nm were individually identified, demonstrating complementary patterns. These images are single frames without any averaging or denoising. Figure 3e shows an overlaid channel of PS and PMMA, clearly illustrating the distribution of each domain. Additionally, an AFM topography image in Supplementary Fig. S3 demonstrates the representative morphology of the phase-separated sample. PS domains appear as irregular-shaped holes with smaller heights, while the higher areas correspond to the PMMA matrix. The height difference between PS and PMMA domains is less than 80 nm, indicating the superior sensitivity of OPT in distinguishing neighboring polymer domains with distinct spectral features. Cross-section line profiles in Fig. 3f further confirm the complementary distribution of the two domains, with PMMA domains smaller than 500 nm in diameter detected within the surrounding PS domains.

OPT microscopy was also utilized to image fabricated nanostructures of homopolymers. As illustrated in Fig. 3g, h, PMMA striped patterns, prepared via electron beam lithography, were imaged at 1170 nm. Each stripe measures 500 nm in width and 650 nm in height, with a distance of 1.5 μm between the centers of neighboring stripes, as confirmed by the AFM topography image. Cross-section profiles of OPT signals and height information along the dashed line were co-plotted in Fig. 3i. A Gaussian-like shape of the OPT signal on each stripe is observed, resulting from the convolution of the illumination laser's point spread function (Gaussian) and the cuboid-shaped sample. Both the PS-PMMA phase separation and PMMA striped structures were examined in ambient air, demonstrating the potential of OPT microscopy for applications in materials science, particularly in the design of thin film devices with tailored properties.

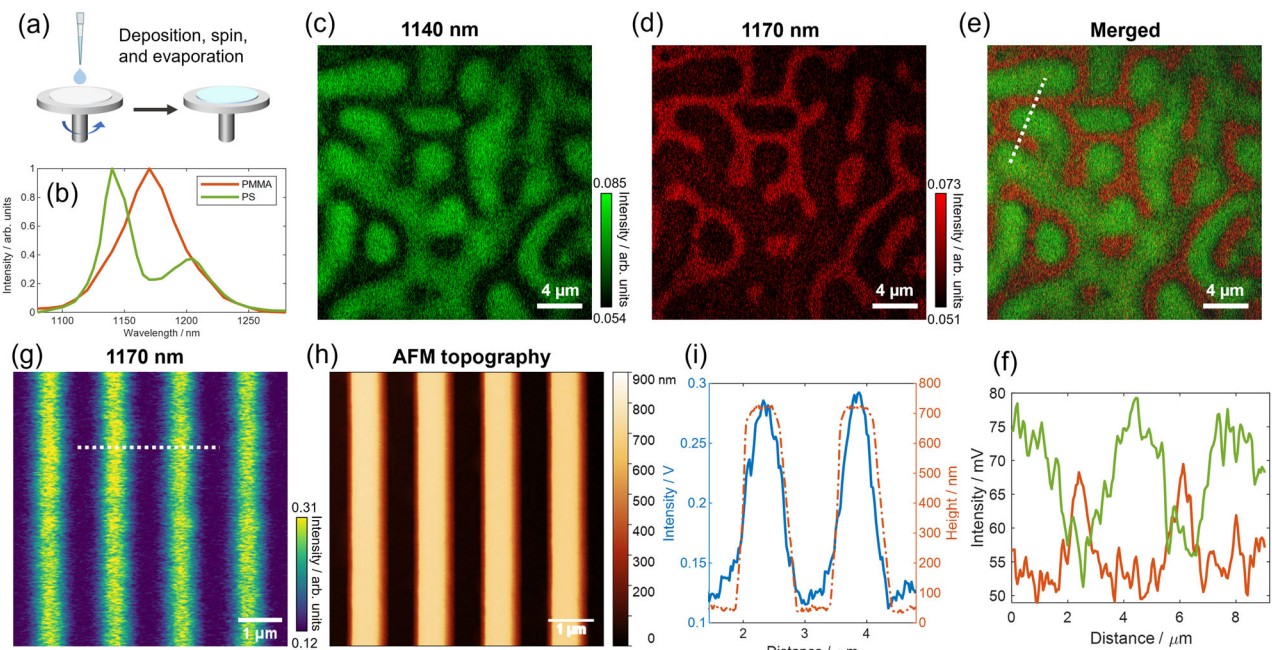

**Fig. 3 | Overtone photothermal (OPT) spectroscopy and imaging of phase-separation patterns of PS & PMMA blends and stripe-structured PMMA.** **a** Schematics of spin-coating procedures used for the preparation of microphase separation patterns of PS and PMMA blends. **b** OPT spectra of pure PMMA and PS as a guide for wavelength selection for PS and PMMA imaging. **c**, **d** OPT single-frame images at 1140 nm and 1170 nm, corresponding to the absorption of PS and PMMA, respectively. These images present complementary patterns of the distribution of PS and PMMA domains. **e** Overlaid channel of **c**, **d**, showing the combined distribution of PS and PMMA domains. **f** Plot of cross-section profiles along the dashed line marked in **e**. **g** OPT image of the striped PMMA nanostructures at 1170 nm. **h** Corresponding AFM topography image of the same field-of-view, providing additional structural information. **i** Co-plot of the OPT signal cross-section and the height information of PMMA stripe nanostructures. **a** was created with BioRender.com released under a Creative Commons Attribution-NonCommercial-NoDerivs 4.0 International license.

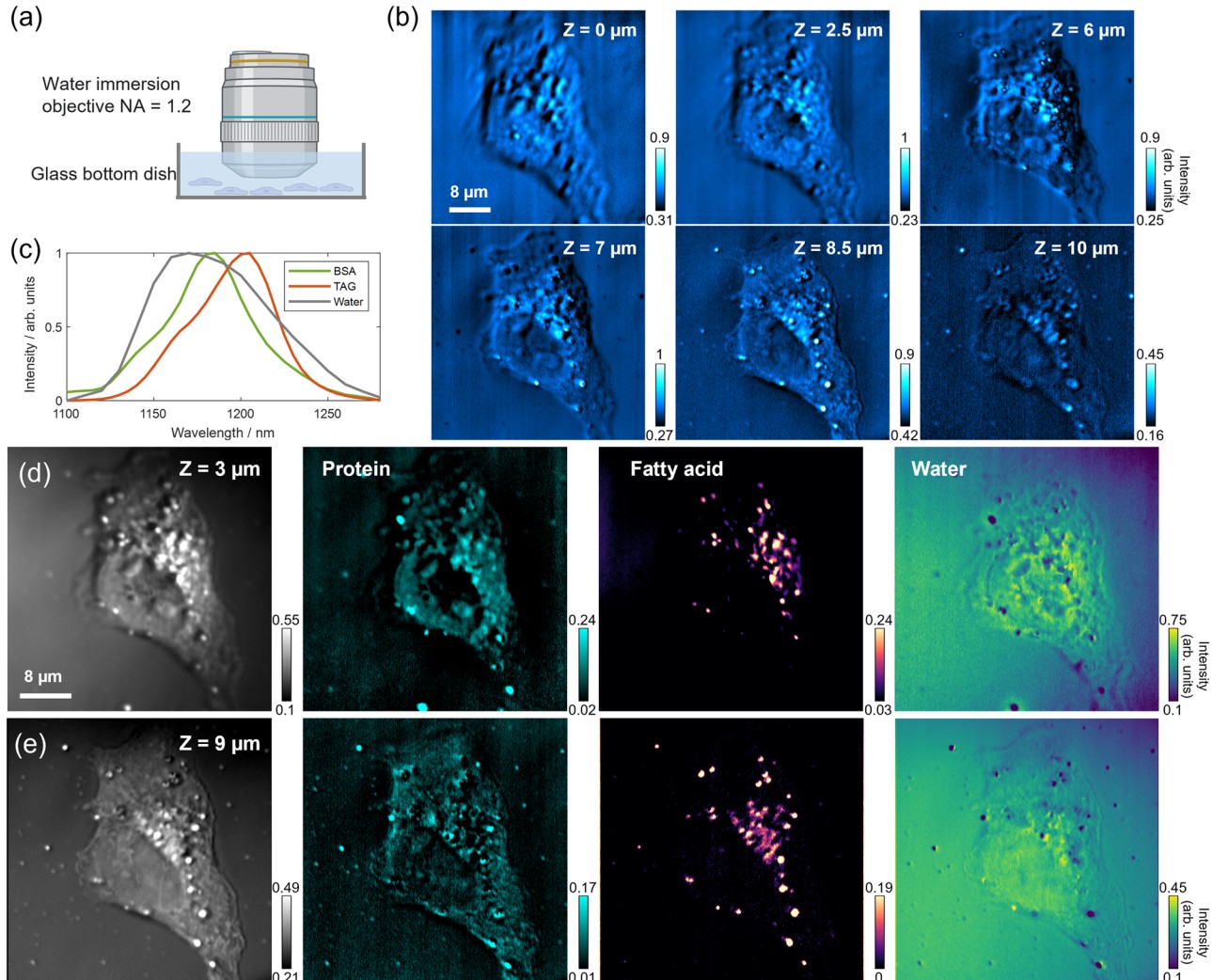

**Fig. 4 | Depth-resolved overtone photothermal (OPT) imaging and spectral unmixing to reveal the spatial distribution of protein and fatty acids in OVCAR-5 single cells. a** Schematics of using a water immersion objective for cell imaging in OPT microscopy. **b** Single-frame OPT image at 1190 nm at multiple Z positions. 1190 nm falls within the range of overlapped absorption of protein and fatty acids. **c** OPT spectra of pure BSA, TAG, and water, serving as spectral references in LASSO spectral unmixing. **d** Raw average OPT image at $Z = 3\,\mu m$ and resulting protein, fatty acid, and water maps. The water background was removed from protein and fatty acid channels through the unmixing of the water component. **e** Raw average OPT image at $Z = 9\,\mu m$ and resulting protein, fatty acid, and water maps. **a** was created with BioRender.com released under a Creative Commons Attribution NonCommercial-NoDerivs 4.0 International license.

## Depth-resolved OPT imaging of metabolites inside ovarian cancer cells

The biological aqueous environment is an essential prerequisite for live cell imaging since it allows for the samples to be examined in their native environment to provide crucial insights into cellular processes and interactions. To evaluate the performance of OPT microscopy in aqueous environment, we used PS beads as a model system. As shown in Supplementary Fig. S4a, we observed a water background in the OPT image of 1-μm PS bead clusters, and the directly extracted PS spectrum was obscured due to the contribution of water. However, by subtracting the water background from the raw PS spectrum, we were able to retrieve a correct PS spectrum (Supplementary Fig. S4b). This highlights the necessity of spectroscopic analysis to resolve chemical compositions in biological studies conducted under aqueous conditions.

Moving forward, we delved into mapping the intracellular metabolites of OVCAR-5 ovarian cancer cells, with a specific focus on the protein and fatty acid content and distribution, as lipogenesis is a biomarker for cancer cells[31,32]. Given the heterogeneity observed in different layers of a single cell, we recognized the need for depth-

sectioning spectroscopic imaging to accurately acquire metabolic distributions. Therefore, we performed depth-resolved hyperspectral OPT imaging of cancer cells at incremental axial slices. Prior to measurement, the cells were fixed in formalin and washed three times in phosphate-buffered saline (PBS) buffer. Figure 4b displays the OPT images at 1190 nm of multiple focal planes of a single cell, and various cell features were observed at different sectioned layers. As the sample stage moved closer to the objective, the imaging plane moved towards the bottom of the dish, allowing us to observe the nucleus and nucleolus more clearly when the axial focus was at the top part of the cell, while lipid droplets were found more distributed at the bottom of the cell. It's important to recognize that the detected OPT signal intensity is influenced by both the scattered light intensity and the modulation depth resulting from photothermal absorption. This characteristic indicates the potential for bias in the determination of absolute chemical concentrations due to optical scattering. In Supplementary Fig. S5, we present transmission images (DC signals) without SWIR beam excitation, alongside OPT images (AC signals) of single cancer cells. This comparison allows for a comprehensive assessment of the potential

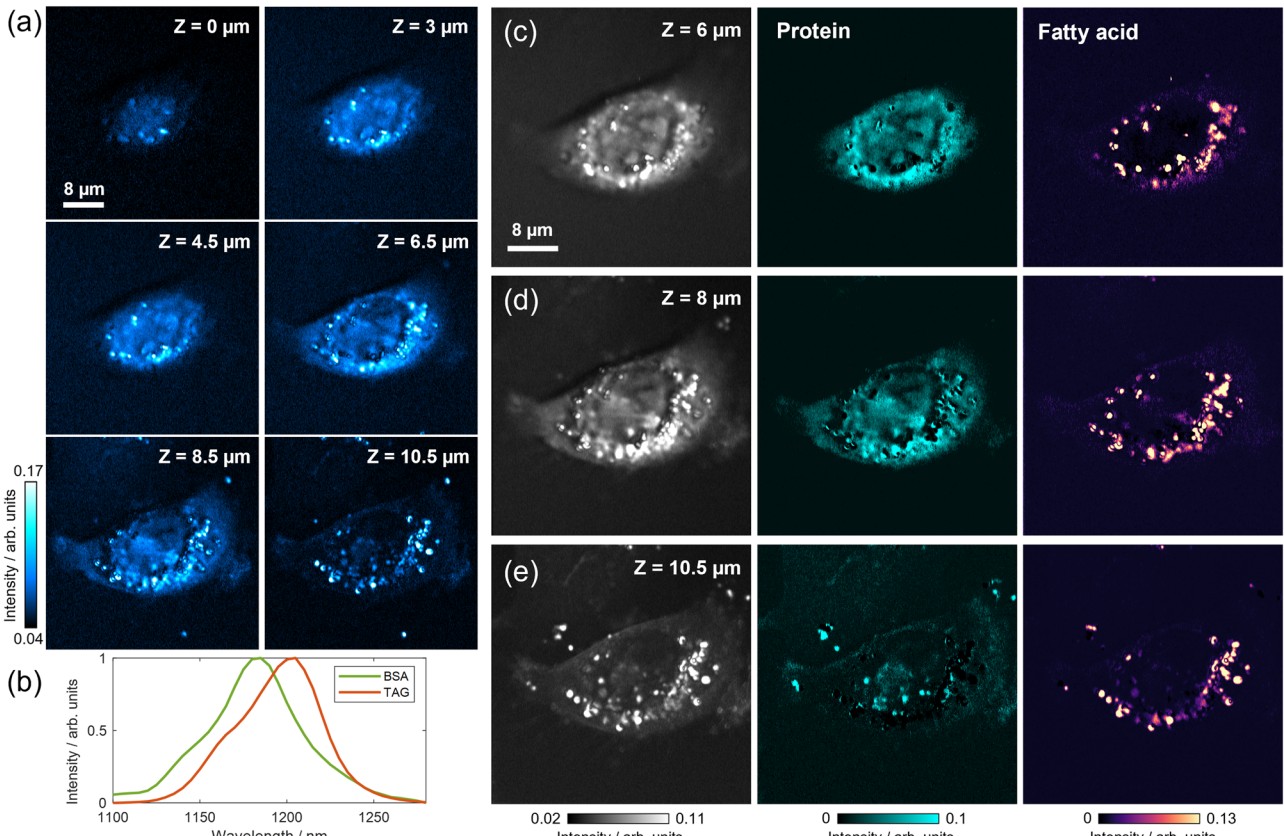

**Fig. 5 | Depth-resolved overtone photothermal (OPT) imaging and spectral unmixing of OVCAR-5 single cells in D₂O PBS. a** Single-frame OPT images at 1190 nm at multiple focal planes showing different cellular regions. **b** Reference OPT spectra of BSA and TAG for LASSO hyperspectral unmixing. **c–e** Raw OPT images and corresponding protein and fatty acid maps at $Z = 6\,\mu m$, $8\,\mu m$, and $10.5\,\mu m$, respectively.

bias. Consequently, the subsequent chemical decomposition via OPT microscopy is carried out in relative concentration (semi-quantitative) measurements, rather than precise absolute quantitative values.

To accurately generate chemical concentration maps of protein and fatty acid, we leveraged the least absolute shrinkage and selection operator (LASSO) algorithm[33,34] for a pixel-wise unmixing based on their spectral profiles. Detailed LASSO algorithms and procedures are described in the Methods Section. Our high spectral resolution allowed us to distinguish between the overlapped peaks of C-H absorption. By employing sparsity-constrained hyperspectral image unmixing via the pixel-wise LASSO algorithm, we were able to further enhance the chemical resolving power of our instrumentation. We collected OPT spectra of bovine serum albumin (BSA) and triglyceride (TAG) and used them as reference chemicals for concentration unmixing of protein and fatty acid components (Fig. 4c)[35]. To exclude the water absorption background and address the overlapped spectral shapes, we performed three-component spectral unmixing for protein, fatty acid, and water. Raw average OPT images and the generated chemical maps of protein, fatty acids, and water at two depths are demonstrated in Fig. 4d, e. Our analysis revealed that protein constitutes a significant component of the cell body, and its OPT signal generally becomes weaker when focusing on the bottom layer of the cell. Conversely, the concentration and distribution of fatty acids, which mainly manifest as lipid droplets, vary significantly at each depth. Besides, lipid droplets contain a heterogeneous protein abundancy due to the presence of proteins or enzymes on the surface that regulate lipid metabolism. Some of the droplets observed in Fig. 4d, e are protein-rich, while a few others are purely dominated by the lipid cores. The inclusion of

water as a component in the spectral unmixing process was advantageous for both removing background and enhancing image accuracy. By assigning all water signals to a separate image, the uneven background and artifacts were effectively removed from protein and lipid channels, and the resulting chemical maps were clearly distinguished and significantly improved with high signal-to-background ratios.

An alternative way to deal with the water backgroud is replacing the medium with heavy water PBS (D₂O PBS). The absorption cross-section of D₂O within the 2nd overtone range is ~1/50 of water[36], resulting in a negligible signal that is easily removable by background subtraction. Consequently, biomass can be directly visualized and two-component LASSO analysis is sufficient to generate lipid and protein maps. Figure 5 illustrates the depth-resolved imaging of single OVCAR-5 cells in D₂O PBS, where the chemical concentration maps of protein and fatty acids were generated at three focal planes targeting different cellular regions. At $Z = 6\,\mu m$, an oval protruding nucleus with nucleoli is observed, surrounded by lipid droplets and aggregates. The main cell body at $Z = 8\,\mu m$ exhibits a protein-rich cell network with embedded lipid droplets. The bottom layer of the cell primarily consists of lipids with minimal protein concentration due to the thin vertical integral volume.

Leveraging the LASSO-based spectral unmixing algorithm, we have achieved depth-resolved intracellular structural and chemical fingerprints within the C-H 2nd overtone absorption window of single cancer cells, as demonstrated in Figs. 4 and 5. High flexibility and applicability of OPT microscopy in biological mapping was demonstrated, regardless of the imaging media employed. This versatility allows for imaging living-cell dynamics in their natural conditions using regular PBS or performing fixed-cell imaging in D₂O

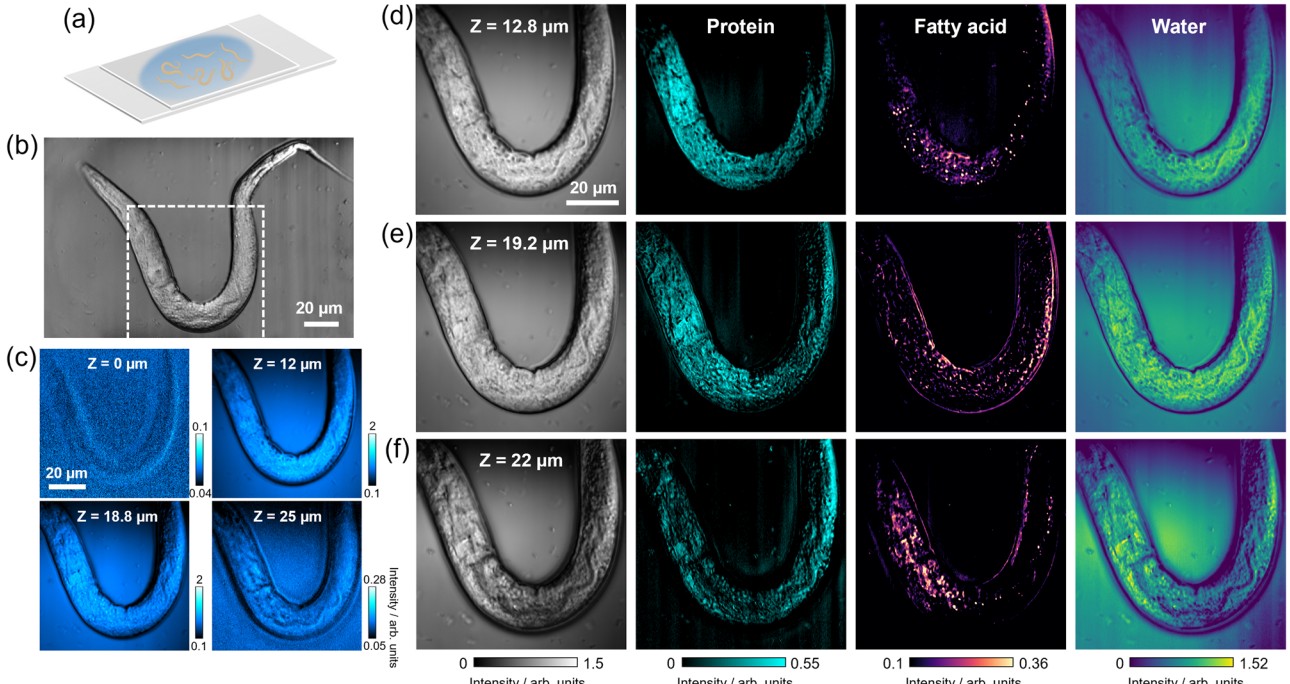

**Fig. 6 | Depth-resolved overtone photothermal (OPT) imaging and spectral unmixing on *C. elegans* worms in aqueous PBS. a** The worms were immersed in PBS and gently sandwiched by two coverslips for imaging. **b** A full field-of-view OPT image at 1190 nm capturing an entire worm, with a zoomed-in area of the body marked for detailed analysis. **c** Single-frame OPT image at 1190 nm of the zoomed-in section of *C. elegans* at $Z = 0$ μm, 12 μm, 18.8 μm, and 25 μm. **d–f** Raw average OPT images and their corresponding protein, fatty acid, and water concentration maps at $Z = 12.8$ μm, 19.2 μm, and 22 μm, respectively. **a** was created with BioRender.com released under a Creative Commons Attribution-NonCommercial-NoDerivs 4.0 International license.

PBS, thereby offering nearly background-free imaging and analysis capabilities.

### Depth-resolved OPT metabolic mapping of *C. elegans*

With 3D spatial resolution, OPT microscopy has prospect in studying larger and more complex biological systems. To explore this potential, we performed in vivo OPT mapping of the multicellular organism *C. elegans*, a model system in biological studies due to its simple anatomy and well-characterized nervous system. The chemical specificity within *C. elegans* provides insights into the mechanisms underlying its behaviors and potential applications in therapeutic medical studies[37].

Figure 6 presents OPT images of multiple layers of a zoomed-in worm body in an aqueous PBS environment. The worm observed was at an early developmental stage, with a diameter of ~20 μm and a length of 200–300 μm. To generate chemical maps, we performed a three-component (protein, fatty acid, water) LASSO unmixing. Despite moderate water absorption in the raw OPT images of aqueous PBS medium, the spectral unmixing successfully removed the water background and generated protein and fatty acid concentration maps. Additionally, we performed OPT hyperspectral imaging in $D_2O$ PBS on a different *C. elegans* worm, as depicted in Supplementary Fig. S6. The two datasets exhibited similar chemical spatial specificity, with protein distributed uniformly throughout the entire worm body while fatty acids predominantly localized near intestinal cells and skin-like hypodermis, which serves as the organism's fat storage sites[38]. We also visualized a large number of native lipid droplets with various sizes and morphology in hypodermal adipocytes. Our sectioning chemical maps revealed the varying 3D distributions of not only protein and lipid granules but also major tissues such as the intestine, muscle, and hypodermis. These findings demonstrate the promising potential of OPT microscopy to provide quantitative insight into chemical information in large and complex

multicellular biological systems, with a flexibility of choosing $H_2O$ PBS or $D_2O$ PBS as the media.

### OPT mapping of lipids and proteins in a highly scattering mouse brain tissue

The brain is a complex system composed of diverse cell types and specialized functional regions. In-depth exploration of local cerebral functions and the determination of in situ chemical compositions are essential for a comprehensive understanding of brain tissue dynamics. Among these functional regions, the corpus callosum holds a pivotal role, serving as the primary white matter tract in the brain by connecting the cortices of the two cerebral hemispheres and facilitating interhemispheric communication[39]. To investigate the applicability of OPT microscopy in this context, we conducted depth-resolved imaging of heterogeneous structures within a mouse brain slice, with a particular focus on the corpus callosum regions. Furthermore, we utilized the LASSO spectral unmixing algorithm to effectively delineate individual protein and fatty acid distributions across two distinct structural layers.

Mouse brain slices, 200 μm in thickness, were prepared and immersed in PBS for subsequent OPT imaging. In Fig. 7a, b, a schematic illustration of the mouse basal forebrain section is presented, along with a transmission image capturing a specific region of the corpus callosum. The image stack, acquired at 1190 nm for varying depths ranging from 0 to 100 μm, substantiates enhanced penetration capabilities compared to MIP measurements, which achieve an approximate depth of 40 μm on brain slices (Supplementary Fig. S7). These depth-resolved images provide a comprehensive view of structural and chemical variations from the surface to deeper layers. Notably, we observed pronounced structural heterogeneity, including sparsely distributed nerve fibers and thick nerve fiber bundles, indicating a complex microarchitecture. To account for potential optical scattering effects, DC (transmission) images at

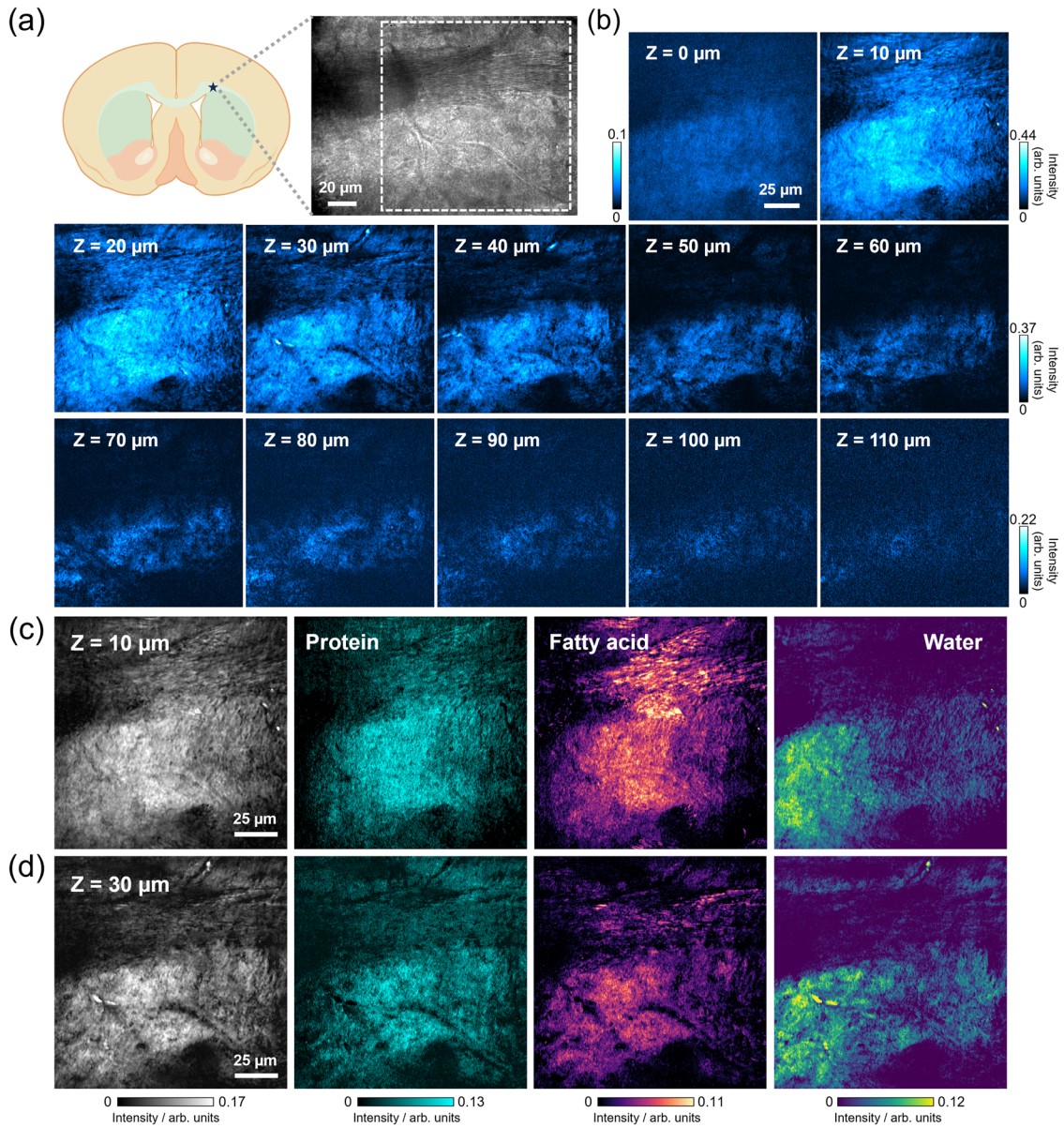

**Fig. 7 | Overtone photothermal (OPT) imaging and spectral unmixing of mouse brain slices. a** Schematic illustration of mouse basal forebrain coronal sections. The green-shaded area highlights the striatum region, where a key component is corpus callosum, marked by a star. A zoomed-in view shows a transmission image captured by an eyepiece camera, focusing on the area of interest within the corpus callosum. The dashed square outlines the region of OPT imaging. **b** Single-frame depth-resolved OPT images at 1190 nm within a depth range of 0–110 μm.

Structural heterogeneity was identified and a penetration depth of ~100 μm can be achieved with OPT microscopy for imaging in turbid media. **c** A raw average OPT image at $Z = 10$ μm and corresponding protein, fatty acid, and water maps. **d** A raw OPT image at $Z = 30$ μm and corresponding protein, fatty acid, and water maps. **a** was created with BioRender.com released under a Creative Commons Attribution-NonCommercial-NoDerivs 4.0 International license.

depths of 10 –100 μm are included in Supplementary Fig. S8 for reference.

It's worth noting that, in the current configuration of OPT microscopy, the penetration depth is primarily determined by the visible probe beam rather than the SWIR beam. The 520 nm beam encounters more pronounced wavefront distortion and focus broadening than the SWIR beam while transmitting through a turbid medium, due to considerable multiple scattering effects. As demonstrated in Supplementary Fig. S9, the SWIR beam exhibits the capability to penetrate through the entire working distance (0.28 mm) of our water objective without substantial beam focus distortion.

Consistent with prior literature, white matter is known to be lipid-rich, and the corpus callosum, in particular, is characterized by a high myelin content, resulting in a substantial lipid presence[40,41]. Our analysis of protein and lipid distribution maps, as illustrated in Fig. 7c, d, supports this observation. Proteins are primarily associated with structural elements, whereas lipids prevail, especially within the relatively sparse nerve fibers. Owing to the high resolution of OPT microscopy, individual fibers in the white matter can be observed. The water content maps are provided simultaneously.

## Discussion

SWIR imaging has great potential for the visualization of biological structures and metabolites deep inside opaque tissues. A panel of SWIR spectroscopy and imaging technologies, such as hyperspectral transmittance and reflectance[8], photoacoustic[14], and OCT (optical

coherence tomography)-based imaging[42], have been widely adopted for their curial roles in biomedical functional and chemical analysis. To accommodate a larger FOV and deeper penetration, current SWIR imaging approaches often sacrifice lateral resolution and detection sensitivity. In this work, we introduced OPT microscopy, utilizing SWIR excitation of overtone bands coupled with a visible beam to probe thermal effects. OPT microscopy achieves both high resolution and high sensitivity, benefiting from its pump-probe detection scheme, the utilization of a visible probe, and a high numerical aperture (NA) objective. As demonstrated in the results of polymer nanoparticles, OPT microscopy offers a lateral resolution of 405 nm and can clearly detect individual 200-nm PMMA beads. This is the first time to achieve simultaneous high spatial resolution and high sensitivity imaging within the SWIR region. We should acknowledge that such advantages come with a compromised imaging depth of ~100 μm. Nevertheless, the enhanced imaging resolution and sensitivity broadens the applicability of SWIR spectroscopy, enabling the visualization of subcellular features, characterization of materials such as nano-plastics or nanoscale organic contaminants in semiconductors, and identification of small drug particles in pharmaceutical products.

Photothermal microscopy is a pump-probe technique that optically detects local thermal gradients with high sensitivity. In photothermal microscopy, molecules or nanoparticles in a sample selectively absorb the pump excitation beam and undergo local temperature increase. The temperature change induces a local variation of the refractive index of the medium, thus altering the transmission or scattering of the probe beam. To date, two main categories of photothermal microscopy have been developed based on the choice of excitation frequency in the visible or mid-infrared. Visible photothermal microscopy leverages the field enhancement of gold nanostructures and targets single light-absorbing nanoparticles, such as single nonfluorescent dye molecules[43] and semiconductor nanocrystals[44]. To enable bond-selective photothermal imaging, MIP microscopy offering abundant chemical information in the fingerprint region has been developed[21,26,45,46]. MIP has been successfully applied to material and life science. Examples include mapping of Fabry–Pérot modes of single metal nanowires[47] and local cation heterogeneities of perovskites[48], probing bacterial metabolic responses at a single-cell level[49–51], structural mapping of proteins in cells and tissues[52,53], and mapping of enzymatic activities through MIP reporters in the silent window[54]. Despite these achievements, MIP microscopy faces a strong absorption of water, resulting in substantial optical losses in thick tissues and a considerable background for measurements in aqueous environments.

In comparison, OPT microscopy provides chemical specificity while circumventing the substantial optical attenuation caused by water absorption in the mid-IR range. This enables versatile imaging applications, from observing living cells in their native environment to achieving deeper penetration into biological tissues. Including hyperspectral unmixing analysis further enhances the chemical resolving power in the SWIR window, where multiple functional groups exhibit overlapping absorption peaks. Supplementary Table 1 summarizes the attributes of current SWIR imaging methods, MIP microscopy, and OPT microscopy.

A key advantage of OPT over MIP microscopy lies in its compatibility with high NA objectives. Existing mid-infrared imaging platforms often employ parabolic mirrors or reflective objectives with an effective NA of up to 0.8. In contrast, the SWIR range is not limited by such optical constraints and can fully exploit high NA objectives, including water-immersion and oil-immersion objectives. This feature significantly increases the focused power density, and boosts signal intensities. Moreover, the optical configuration of co-propagation-based laser scanning of SWIR and probe beams simplifies instrumentation complexity and streamlines alignment procedures. This simplicity stands in contrast to mid-IR photothermal microscopy,

which compromises lateral resolution through co-propagation via a low-NA objective. To maintain high lateral resolution in mid-IR approaches, a counter-propagation configuration of the mid-IR beam and probe beam is typically required, resulting in more challenging alignment and laser scanning implementation.

One limitation of OPT microscopy lies in the strong overlap of overtone bands in the SWIR spectroscopic window, which results in crosstalks between chemical species. In this work, we sought to improve the specificity through synergistic innovations in instrumentation and data science. Compared with traditional SWIR technologies, OPT can perform hyperspectral imaging at a subcellular resolution of 405 nm. We reason that under such resolutions, for a spatially heterogeneous biological system, only a few species have dominant contributions within each laser scanning spot. This physical prior knowledge, which we refer to as "local chemical sparsity", is mathematically translated as an L1-norm regularizer on the concentration vector at each pixel (i.e., pixel-wise LASSO), thereby suppressing signal crosstalks effectively. Nevertheless, several challenges remain in advancing the quantitative multiplexing capability of OPT spectroscopic imaging. First, photothermal lensing detection is known to scale with both absorption and local optical scattering, the latter of which can result in biased concentrations after unmixing. This issue could be potentially addressed by either leveraging the water unmixing channel as a local calibration metric to compensate for scattering bias, or through quantitative phase detection, which scales solely with refractive index. Second, the broad spectral peaks in the SWIR region still pose a high demand for the SNR of the data. Heuristically, the necessary SNR can be determined by inspecting the quality of the output concentration maps. With sufficient SNR, maps of different chemical species should reflect the distinct morphological structures of subcellular organelles that differ from the water background channel, such as lipid droplets in the fatty acid channel. Lastly, to capture all the possible spectral differences amidst broad spectral peaks, we measured the entire second overtone window permitted by the laser tuning range. This approach is not optimized, as the number of separable species is much fewer than the spectral frames and theoretically requires fewer spectral frames. In the future, we aim to perform recursive feature elimination to selectively measure important spectral bands that contribute most to the differentiation of species.

Biological tissues, including brain slices, are inherently turbid media, complicating 3D volumetric rendering via one-photon multiple scattering[55,56]. Given that the pump-probe scheme inherently constitutes a "two-photon" imaging process, OPT microscopy demonstrates superior sectioning capability with insensitivity to the scattering loss of either the pump or probe beam, which does not contribute to the actual OPT signal. This unique characteristic effectively mitigates background from one-photon scattering and underscores the promising potential of OPT microscopy for bond-selective imaging through inhomogeneous turbid matrices. Taking a step further, leveraging a probe beam with longer wavelengths will enhance the performance for deeper penetration in turbid species, while maintaining excellent sectioning capability.

The OPT microscope, based on an ultrafast near-infrared laser, can be easily adapted into a multimodal imaging platform. Various imaging techniques, such as two-photon fluorescence, transient absorption, and coherent Raman scattering, can all be readily implemented on this platform with a simple exchange of photodetectors and electronics for signal extraction. This creates a highly versatile integrated multifunctional imaging station suitable for material and biological research in a wide span. Another notable feature of OPT is its utilization of a frequency-doubled 520 nm as probe beam, allowing for improved sensitivity and spatial resolution in detecting temperature change. Additionally, the probe beam exhibits a low noise level (near shot-noise-limited) which is attributed to the quiet ultrafast beam

outputs of our laser. By employing a second beam as the photothermal detector, the OPT microscope benefits from the fact that the detection noise is largely determined by the noise floor of the probe beam, thereby relaxing the criteria for the noise level of the excitation beams. Consequently, this characteristic opens up possibilities for incorporating different laser sources, even those with noisier outputs, into the OPT microscope, which may potentially facilitate multiple promising technical advancements such as cost reduction, expanding the spectral coverage range, and enhancing wavelength tuning speed.

Building upon the less stringent requirements for excitation lasers, various enhancements to the capabilities of OPT microscopy can be explored. Firstly, our proof-of-principle implementation of OPT microscopy focused on C-H bonds within a limited portion of SWIR range. Expanding the wavelength coverage by employing a broadband laser, such as a supercontinuum white light laser, would extend OPT into the 1$^{st}$ overtone region. This expansion would enable investigations targeting various functional groups and meanwhile offer a cost-effective approach for OPT measurements. Such advancements would be particularly beneficial for pharmaceutical investigations. Addressing the imaging speed aspect, the current acquisition time for a 400 pixel × 400 pixel OPT image is ~3 s, while manual wavelength tuning takes around 5 s, significantly compromising the speed of hyperspectral imaging. Incorporating an automatic and fast wavelength tuning module would enhance the overall imaging speed, thus creating opportunities for dynamics studies of living cells.

Another prospective implementation for broadening the scope of OPT microscopy is the incorporation of epi detection. This will be particularly useful when dealing with the measurement of small nanoparticles with intense backscattering, as well as opaque samples like pharmaceutical tablets. Our future work aims to augment the existing transmission mode by introducing an add-on epi detection mode, thereby enabling a wider range of applications.

In summary, we have developed overtone photothermal (OPT) microscopy, a label-free bond-selective imaging approach that significantly improves the sensitivity and lateral resolution of current SWIR image modalities. OPT fills the gap between visible and mid-infrared photothermal microscopy by enabling highly chemical-specific mapping in aqueous environment. We demonstrated successful applications of OPT microscopy in mapping polymer nanostructures and performing depth-resolved spectroscopic imaging and metabolic profiling of intercellular and multicellular organisms at single-cell and tissue levels. OPT microscopy allows for the unambiguous identification of spatially dependent protein and lipid distributions. We anticipate that the integration of OPT microscopy with existing large-scale SWIR imaging techniques will facilitate the exploration of structural and chemical features across various imaging scales, providing valuable insights into both material science and the underlying mechanisms of complex biological systems.

## Methods

### OPT microscope

OPT microscopy employed an InSight DeepSee femtosecond laser (Spectra-Physics), which emits a tunable beam from 680 nm to 1300 nm and a fixed-wavelength beam of 1040 nm with a repetition rate of 80 MHz. The OPT experiments were conducted within the 1080 nm to 1280 nm range, corresponding to the 2nd overtone resonances of C-H stretching vibrations. The power of two laser outputs was adjusted using a combination of a half-wave plate and a polarization beam splitter. An acousto-optic modulator (AOM, M1205-P80L-0.5, Isomet) was utilized to modulate the excitation beam at desired repetition rates (100 kHz–1 MHz) and duty cycles (usually 50% for biological measurements). The SWIR beam then passed through five 15-cm SF57 glass rods for chirping purposes, with a chirped pulse duration of approximately 2 ps to minimize nonlinear damage. The 1040 nm output from the laser source was frequency-doubled using a

lithium triborate (LBO) crystal (LBO-604H, Eksma Optics) to 520 nm and served as the probe beam. Similarly, the femtosecond 520 nm probe beam was chirped after passing through four 15-cm glass rods to reduce non-linear thermal damage.

To optimize the OPT signal, one lens pair on the probe beam path was used to adjust the beam convergence and divergence, controlling the axial displacement between the excitation beam and probe beam foci. The SWIR beam and 520 nm probe beam were combined using a 900 nm long-pass dichroic beam splitter (DMLP900, Thorlabs) and directed to 2D scanning Galvo mirrors. The samples were mounted on an upright microscope (BX51WI, Olympus), and the images were acquired through laser raster scanning.

During the OPT imaging, the average on-sample SWIR power remained below 50 mW, and the average on-sample power of 520 nm was below 20 mW. For tight focusing of the SWIR and probe beams, a 60X water immersion objective with a NA = 1.2 (UPlanApo/IR, Olympus) was utilized. The collection NA of the oil condenser was adjustable to optimize the OPT signal intensity. After passing through the condenser, the probe beam went through a lens and optical filters before being detected by a biased Si photodiode (DET100A2, Thorlabs). The electronic signal from the photodiode was transmitted through high- and low-pass electrical filters, amplified by a 46 dB low-noise pre-amplifier (SA-230F5, NF Corporation), and then processed by a lock-in amplifier (HF2LI, Zurich Instruments). The demodulated signals from the lock-in amplifier were registered as OPT signals and synchronized with each laser scanning step to generate OPT images. Spectroscopic information was extracted from hyperspectral image stacks acquired by manually tuning SWIR wavelengths and recording OPT images.

### Spectral unmixing with chemical sparsity constraint via pixel-wise LASSO

The task of spectral unmixing is to decompose the high-dimensional hyperspectral image into the multiplication of two lower-dimensional matrices, namely reference spectra of pure chemicals and concentrations, to facilitate comprehension of the chemical information buried within the raw images. Mathematically, we define the spatial dimensions and the spectral dimension as $N_x, N_y$ and $N_\lambda$, and define the number of pure chemicals as $K$, the spectral unmixing problem can be formulated as follows:

$$D = CS + E, \qquad (1)$$

where $D \in \mathbb{R}^{N_x N_y \times N_\lambda}$ is a raster-ordered raw data matrix, $S \in \mathbb{R}^{K \times N_\lambda}$ contains the reference spectra of all pure chemicals, $C \in \mathbb{R}^{N_x N_y \times K}$ includes the concentrations for all the components and $E \in \mathbb{R}^{N_x N_y \times N_\lambda}$ is the residual term containing the fitting error and measurement noise. Given reference spectra of pure chemicals, the task of spectral unmixing is simplified to finding the concentrations at each spatial position. We introduce a L1-norm local chemical sparsity constraint to the concentration vector at each pixel, to encourage that at each position, only few components have dominant contributions. The solution is found by solving for the following LASSO problem in a pixel-wise manner:

$$\hat{C}_i = \arg\min_{C_i} \left\{ \frac{1}{2} \|D(i, :) - C_i S\|^2 + \beta \|C_i\|_1 \right\}, \qquad (2)$$

where $i$ represents the pixel number, $\hat{C}_i \in \mathbb{R}^K$ is the concentration vector for all components at pixel $i$, and $\beta$ controls the level of sparsity at each pixel to ensure suppression of channel cross talks while avoiding over-suppression that leads to all zeros. The value of $\beta$ is fixed across all images to ensure output values can be quantitatively compared.

In the demonstrated spectral unmixing of biological specimens in water environment, OPT spectra of bovine serum albumin (BSA, purity

≥96%, Sigma-Aldrich), triglyceride (TAG, purity ≥97%, Sigma-Aldrich), and deionized water were collected, serving as the pure spectral references for protein, fatty acids, and water, respectively.

## Collection of NIR absorption spectra via UV-Vis-NIR spectrophotometer

Standard NIR absorption spectra for both polystyrene and ethylene glycol were acquired utilizing the Cary 5000 UV-Vis-NIR Spectrophotometer (Agilent) following established protocols. Polystyrene ($M_w$ = 35,000, Sigma-Aldrich) was dissolved in toluene, and 3 mL of the resulting solution was transferred into a quartz cuvette for insertion into the spectrophotometer. An absorption spectrum of toluene was acquired and subsequently utilized for baseline correction. Similarly, 3 mL ethylene glycol (purity 99.8%, Sigma-Aldrich) was introduced into a separate quartz cuvette, with ambient air as the reference baseline. The measurement parameters include a spectral bandwidth of 2 nm, an integration time of 0.1 s, and a wavelength interval of 1 nm.

## Preparation of phase separation patterns of PS and PMMA blends

To prepare the polymer stock solution, 51.4 mg PS ($M_w$ = 35,000, Sigma-Aldrich) and 15.4 mg PMMA ($M_w$ = 35,000, Sigma-Aldrich) were dissolved in 2 mL toluene and allowed to settle. To observe the phase separation pattern of PS-PMMA blends, a coverslip substrate with a thickness of 130 μm was cleaned with acetone and isopropanol, followed by drying using nitrogen gas. A 40 μL mixture solution was spin-coated onto the coverslip substrate using a spinner (Headway Research, CB-15 & PWM32) at 500 revolutions per minute (rpm) for 10 seconds, followed by 1000 rpm for 50 seconds.

## Fabrication of PMMA striped structures

The fabrication of PMMA stripes was achieved using electron-beam lithography (EBL) on a 130 μm thick coverslip substrate. Prior to fabrication, the substrate underwent a thorough cleaning process involving acetone and isopropanol, nitrogen drying, and baking at 110 °C. A 950-PMMA 6% in anisole solution was spin-coated onto the substrate to achieve a PMMA thickness of 600 nm as per the recipe used. The sample was subjected to a hard bake at 180 °C before a conduction layer of Au nanoparticles was sputtered onto the top surface. EBL was performed at 30 keV after a dose test to pattern the sample. The fabrication process was completed by soaking the PMMA in a solution of methyl isobutyl ketone/isopropanol (MIBK/IPA) 1:3 for 70 s, followed by rinsing with IPA and a wet etch to remove the sputtered Au.

## OVCAR-5 cancer cells

The OVCAR-5 cells used in this experiment were a generous gift from Dr. Daniela Matei of Northwestern University. The cells were cultured in RPMI 1640 media (Gibco) supplemented with 10% fetal bovine serum (Gibco), 2 mM L-Glutamine, and 1% penicillin-streptomycin. The cell line was incubated at 37 °C with 5% $CO_2$. For imaging purposes, the cells were seeded at a density of 0.2 million per dish on glass-bottom dishes and fixed in 10% neutral buffered formalin (Sigma-Aldrich) after 24 h of culturing. Prior to imaging, the cells were washed three times with either $H_2O$ PBS or $D_2O$ PBS. OPT images were obtained by directly immersing the objective into the medium in the glass-bottom dish.

## *C. elegans* preparation

*C. elegans* N2 wild-type isolates were purchased from the Caenorhabditis Genetics Center (CGC). The worms were maintained in a 20 °C incubator and propagated on Nematode Growth Medium (NGM) agar plates supplemented with the auxotrophic *Escherichia coli* mutant strain OP50. To immobilize worms for imaging, worms were harvested and washed using phosphate-buffered saline (PBS) solution (Gibco) and then fixed by 10% neutral buffered formalin solution (Sigma-

Aldrich). The *C. elegans* worms were washed using $D_2O$ PBS or $H_2O$ PBS three times and then carefully sandwiched between two cover glasses for imaging. Throughout the measurement process, PBS media was gently added along the edge of the top coverslip to maintain hydration.

## Mouse brain tissue preparation

The brain tissue utilized in this work was from an adult C57BL/6 J mouse. The mouse was sacrificed and subsequently perfused with PBS solution (1X, Thermo Fisher Scientific). The brain was carefully extracted and immersed in 10% formalin solution for a fixation period of 48 hours. Following fixation, an oscillating tissue slicer (OTS-4500, Electron Microscopy Sciences) was employed to section the brain into 200-μm-thick coronal slices. The brain slices were washed three times with PBS solution before measurement. For OPT imaging, the brain tissue was submerged in PBS and sandwiched between two cover glasses, while for mid-IR imaging, it was positioned between a coverslip and a $CaF_2$ substate.

## MIP imaging of brain slices

A tunable quantum-cascade laser (QCL, MIRcat-2400, Daylight Solutions) was utilized to effectively excite the vibrational modes within the sample. Simultaneously, a continuous-wave laser of 532 nm (Samba 532 nm, Cobolt) was used to probe the local thermal changes. The mid-IR beam operated at a repetition rate of 200 kHz, with a pulse width of 500 ns. The average mid-IR power applied to the sample was approximately 3 mW, while the probe power was set at 45 mW. The two beams were collinearly focused onto the sample plane through a reflective objective, and an objective Z-piezo stage controlled the axial position to acquire multi-depth images. The forward propagating probe photons were detected by a biased photodiode (DET100A2, Thorlabs). The photocurrent generated by the photodiode underwent amplification through a low-noise amplifier (SA-251F6, NF Corporation) and was subsequently directed to a lock-in amplifier (HF2LI, Zurich Instruments) for signal demodulation. A multichannel data acquisition card registers the data for real-time signal processing.

## Data availability

All data supporting the findings of this study are available within the main article and the Supplementary Information file. The raw data are available from the corresponding author upon request.

## Code availability

MATLAB code for LASSO spectral unmixing is available on the following website: https://github.com/buchenglab.

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

## Acknowledgements

This work was supported by NIH grants R35GM136223, R33CA261726, R01EB032391, and R01EB035429 to J.X.C.

## Author contributions

J.X.C. generated the idea and guided the overall research; L.W. built the imaging flatform, performed OPT data collection, and analyzed all data; H.L. conceived the idea of introducing SHG probe beam and developed pixel-wise LASSO spectral unmixing algorithm; Y.Z. and X.G. conducted the initial explorations of $3^{rd}$ overtone absorption; M.L. prepared brain slices and collected mid-infrared photothermal images; J.L. fabricated the PMMA striped nanostructures; F.C. provided the OVCAR-5 cancer cells; M.Z. provided the *C. elegans* worms. L.W. prepared the manuscript with input from the authors.

## Competing interests

The authors declare no competing interests.
