## [Peer Review File · Nature Communications]

Overtone photothermal microscopy for high-resolution and high-sensitivity vibrational imagingREVIEWER COMMENTS

Reviewer #1 (Remarks to the Author):

High-resolution and high-sensitivity vibrational imaging via overtone photothermal microscopy
Summary

The authors developed a technique utilizing overtone photothermal microscopy with excitation in the SWIR region for imaging to provide high resolution and high sensitivity. The paper details the method employed along with a demonstration of its capability for chemical imaging of various biological systems/organisms. The approach described has significant potential to address key practical challenges arising in prior implementations of O-PTIR in the mid-infrared, particularly for analysis of turbid species. Overall, the work is novel and interesting, and worthy of consideration for publication in Nature Communications. The manuscript is recommended for publication following minor revisions summarized below.

1. The authors compare and contrast their method with diffuse optical spectroscopic imaging and photoacoustic microscopy, but interestingly not hyperspectral reflectance or transmittance imaging. Given the ubiquity of NIR spectroscopic imaging tools available, it would seem natural to compare absorbance measured from extinction+scattering of the transmitted NIR with absorbance determined by O-PTIR. The authors are encouraged to include a discussion of more conventional hyperspectral imaging with NIR sources for comparison and contrast in the Introduction.
2. The authors are encouraged to better emphasize the significance of the common-path optics, in which both pump and probe beams copropagate through a galvanometer mirror pair for beam-scanning through a 4F relay lens pair to a refractive objective. The simplicity of this configuration relative to the requirements to copropagate mid-IR and visible greatly expands the scope of use of the method in a manner not sufficiently emphasized in the current draft.
3. The authors are also further encouraged to add a discussion to the manuscript on the potential applications of the method for vibration-selective imaging in turbid media. The pump-probe configuration shown in Figure 1 produces a signal that scales with the product of the pump and probe intensities. As such, it is expected to share the same inherent confocal sectioning capabilities as two-photon excited fluorescence, with corresponding insensitivities to scattering losses in either the pump or probe that do not contribute to probe-beam modulation. Although beyond the scope of the present work, the potential for enabling vibration-selective microscopy in 3D volume renderings within turbid matrices is quite exciting.
4. The authors point out (correctly) the need for an axial offset between the pump and probe beam in a copropagating geometry to maximize the sensitivity of the probe beam perturbation from pump-beam local heating. However, the manuscript would be strengthened by a bit more exposition on the underlying mechanism for those not as intimately familiar with the intricacies of photothermal microscopy.
5. The OPTIR measurements have the potential to be biased by optical scattering if reported solely in terms of the detected modulation amplitude from the lock-in amplifier. Amplitude scales both with the total scattered signal and the depth of modulation from absorption, preferentially highlighting edges through mechanisms akin to dark-field imaging. Normalization of the F-PTIR signals by the DC scattered intensity should recover depth of modulation values with more direct connections back to absorption alone. The authors are encouraged to either report depth of modulation measurements (rather than amplitude of modulation), or also include images of the DC signal amplitude measured simultaneously to aid in assessing potential bias contributions from optical scattering. The results shown in Figure 4 come directly to mind.
6. While the authors are certainly correct in concluding that overtone and combination bands exhibit reduced attenuation of the pump and probe beams from lower absorption cross-sections, that reduced cross-section affects not only the background but the signal as well. Reducing both the signal and the noise proportionally does not improve the signal to noise ratio. The manuscript should be updated to include a more thorough discussion of the conditions for which major SNR enhancements are anticipated to arise (if at all) relative to O-PTIR with mid-infrared excitation. The results shown in Figure 3 suggest relatively low SNR and spatial resolution compared to previously reported O-PTIR measurements
7. The spectroscopic window accessible by NIR is dominated by overtones and combination bands

incorporating H-stretching motions. This window historically lacks high selectivity, consistent with the highly overlapping peak shapes shown in Figure 4C. The authors are encouraged to include a greater discussion of the limitations associated with decomposing signals from overlapping spectral features, particularly if image contrast scales not just with absorption alone but also with local optical scattering. Specifically, what level of SNR is necessary and over what spectral range for confidently assigning composition on a per-pixel or per-segment basis?

Reviewer #2 (Remarks to the Author):

The authors presented an interesting approach that suggests minimizing the water signal using SWIR in a photothermal approach. They argued that previously published approaches utilizing visible and FTIR methods have significant limitations. The novelty lies in combining the photothermal approach with hyperspectral SWIR. However, the manuscript fails to acknowledge that hyperspectral SWIR has been used for almost a decade to minimize water and scattering in deep tissue, which is a major limitation. It remains unclear whether adding photothermal to the currently used hyperspectral SWIR provides significant value.

Overall, the manuscript is technically solid, and the idea of combining photothermal with SWIR is novel, potentially acceptable if the authors demonstrate the advantages directly. That is the major issue. Several minor technical issues also need to be addressed:

1. The authors should mention that the SWIR region is not uniform, and large penetration depth is only possible at certain wavelengths. They can refer to existing literature that measures depth versus each wavelength. However, for cell samples, this might not be an issue.
2. Please explain why the LASSO method was chosen for unmixing and its advantages over other techniques.
3. Washing cells with D₂O to minimize water absorption might lead to cell death, precluding live cell imaging. Please comment on this concern, as the same might apply to *C. elegans*.
4. The methods section should include more technical details, such as the source of albumin, triglycerides, and specifics of the NIR/VIS/SWIR spectrophotometer.
5. In Figure 2, "spectral fidelity" usually refers to spectral calibration. Please check if this term is correct in the given context.
6. The shape of the spectra shown by OPT differs somewhat from those previously published by Cao et al. *Journal of biomedical optics* 18.10 (2013): 101318.

Reviewer #3 (Remarks to the Author):

This paper presents a version of photothermal (PT) microscopy that uses overtone vibrations as the primary source of contrast. This work is a PT implementation of a similar idea pursued earlier by the same group, which was based on overtone excitation with photoacoustic (PA) detection. The PT technique itself, based on the excitation of fundamental molecular vibrations (not the overtones), has been a focus of the group for the last 8 years or so. Here the authors claim that overtone PT (OPT) represents several advantages compared to their mid-IR PT work: 1) reduced background from water and 2) higher resolution because better (refractive) lenses can be used.

The paper is interesting and the data shown that the technique has merit. In a sense, it is a new variation of the PT family of techniques. Although certainly new, some of the claims in this work come across as rather strong. In addition, the following points are noted:

- 1) While water absorption in the mid-IR range is substantial, it is also present in the NIR. In addition, the fundamental absorptions in the mid-IR are strong, whereas the 2nd overtones in the NIR are comparatively weak. This means that the weaker (and broadened) NIR features appear on a water background, which also reduces contrast. Therefore, it is not sufficient to state that OPT has an automatic advantage over mid-IR without some sort of quantification in terms of contrast

etc.

2) The images in Figure 4 reveal a substantial water background. In addition, the "protein" image appears to be affected by transmission effects, which goes beyond the more quantitative chemical contrast. These issues are not discussed clearly in the manuscript.

3) "These findings demonstrate the promising potential of OPT microscopy to provide unambiguous and quantitative chemical information in large and complex multicellular biological systems, irrespective of the imaging media used." Too strong of a statement, as the medium here is actually D₂O, which was artificially introduced to suppress the background from water.

4) The *C. elegans* demonstration is nice, but this specimen is very small and thin, which means that the reader still cannot determine the OPT advantage over mid-IR imaging in aqueous samples. In the mid-IR, samples can still be imaged at depths up to ~20 μm , so the current demonstration does not necessarily demonstrate an advantage in terms of deeper imaging in tissues.

Reviewer #1 (Remarks to the Author):

The authors developed a technique utilizing overtone photothermal microscopy with excitation in the SWIR region for imaging to provide high resolution and high sensitivity. The paper details the method employed along with a demonstration of its capability for chemical imaging of various biological systems/organisms. The approach described has significant potential to address key practical challenges arising in prior implementations of O-PTIR in the mid-infrared, particularly for the analysis of turbid species. Overall, the work is novel and interesting and worthy of consideration for publication in Nature Communications. The manuscript is recommended for publication following minor revisions summarized below.

We would like to thank the referee for their thorough review of our manuscript and the constructive suggestions. Please see below for the actions we have taken to address the comments.

1. The authors compare and contrast their method with diffuse optical spectroscopic imaging and photoacoustic microscopy, but interestingly not hyperspectral reflectance or transmittance imaging. Given the ubiquity of NIR spectroscopic imaging tools available, it would seem natural to compare absorbance measured from extinction + scattering of the transmitted NIR with absorbance determined by O-PTIR. The authors are encouraged to include a discussion of more conventional hyperspectral imaging with NIR sources for comparison and contrast in the Introduction.

Reply: We would like to thank the reviewer for this suggestion. In the revised Introduction section, we have now included a brief discussion of conventional hyperspectral imaging approaches. The context is as follows:

“Several SWIR imaging methods have been developed, including hyperspectral reflectance/transmittance imaging, diffuse optical spectroscopic imaging (DOSI), and photoacoustic microscopy (PAM). SWIR hyperspectral imaging, which measures reflectance or transmittance, primarily serves in the qualitative spectral characterization of samples at the macroscale, focusing on properties related to absorption and scattering.²¹⁻²³ Operating in a scanning mode, SWIR hyperspectral imaging utilizes a spectral filter and a camera to construct narrowband hyperspectral images. The resulting resolution is typically determined by the camera's pixel pitch, generally in the range of tens of micrometers.”

2. The authors are encouraged to better emphasize the significance of the common-path optics, in which both pump and probe beams co-propagate through a galvanometer mirror pair for beam-scanning through a 4f relay lens pair to a refractive objective. The simplicity of this configuration relative to the requirements to co-propagate mid-IR and visible greatly expands the scope of use of the method in a manner not sufficiently emphasized in the current draft.

Reply: We appreciate the reviewer for acknowledging our optical configuration and providing this valuable suggestion. We have incorporated the following text in the first paragraph of the Discussion section, addressing the optical configuration:

“Moreover, the optical configuration of co-propagation-based laser scanning of SWIR and probe beams simplifies instrumentation complexity and streamlines alignment procedures. This simplicity stands in contrast to mid-IR photothermal microscopy, which compromises lateral resolution through co-propagation via a low-NA objective. To maintain high lateral resolution in mid-IR approaches, a counter-propagation configuration of the mid-IR beam and probe beam is typically required, resulting in more challenging alignment and laser scanning implementation.”

3. The authors are also further encouraged to add a discussion to the manuscript on the potential applications of the method for vibration-selective imaging in turbid media. The pump-probe configuration shown in Figure 1 produces a signal that scales with the product of the pump and probe intensities. As such, it is expected to share the same inherent confocal sectioning capabilities as two-photon excited fluorescence, with corresponding insensitivities to scattering losses in either the pump or probe that do not contribute to probe-beam modulation. Although beyond the scope of the present work, the potential for enabling vibration-selective microscopy in 3D volume renderings within turbid matrices is quite exciting.

Reply: We are grateful for this valuable input from the reviewer. We agree with the reviewer regarding the advantageous "two-photon" process of photothermal microscopy, which enables good sectioning capability and aids in reducing background from one-photon scattering. This characteristic presents promising prospects for applications in bond-selective imaging within turbid media. Therefore, we conducted multi-depth imaging in 200- μm -thick mouse brain slices, a representative of a highly scattering turbid medium. The results are presented in Fig. 7. We also added one paragraph discussing this advantage in the Discussion section.

“Biological tissues, including brain slices, are inherently turbid media, complicating 3D volumetric rendering via one-photon multiple scattering.^{53,54} Given that the pump-probe scheme inherently constitutes a "two-photon" imaging process, OPT microscopy demonstrates superior sectioning capability with insensitivity to the scattering loss of either the pump or probe beam, which does not contribute to the actual OPT signal. This unique characteristic effectively mitigates background from one-photon scattering and underscores the promising potential of OPT microscopy for bond-selective imaging through inhomogeneous turbid matrices. Taking a step further, leveraging a probe beam with longer wavelengths will enhance the performance for deeper penetration in turbid species, while maintaining excellent sectioning capability.”

4. The authors point out (correctly) the need for an axial offset between the pump and probe beam in a copropagating geometry to maximize the sensitivity of the probe beam perturbation from pump-beam local heating. However, the manuscript would be strengthened by a bit more exposition on the underlying mechanism for those not as intimately familiar with the intricacies of photothermal microscopy.

Reply: Thanks to the reviewer for providing this suggestion. In the updated manuscript, we have revised and included the following text to further elaborate on the mechanism of the axial offset:

“Upon the SWIR illumination, the sample’s selective absorption within the focal volume leads to a local temperature rise and a thermal gradient, inducing a subtle decrease in the local refractive index at the pump beam focus. To efficiently generate photothermal signals, an axial focus

displacement between the two beams is implemented, as depicted in Fig. 1b.³¹ In this scenario, the induced thermal lens modifies the propagation of the 520 nm probe beam, causing it to diverge or converge depending on its focal position relative to the pump focus. When there is no axial offset, the thermal lens forms precisely at the focal position of the probe beam. Therefore, the ray locus of the probe beam will not be modified, and the photothermal signal vanishes.”

5. The O-PTIR measurements have the potential to be biased by optical scattering if reported solely in terms of the detected modulation amplitude from the lock-in amplifier. Amplitude scales both with the total scattered signal and the depth of modulation from absorption, preferentially highlighting edges through mechanisms akin to dark-field imaging. Normalization of the O-PTIR signals by the DC scattered intensity should recover the depth of modulation values with more direct connections back to absorption alone. The authors are encouraged to either report depth of modulation measurements (rather than amplitude of modulation), or also include images of the DC signal amplitude measured simultaneously to aid in assessing potential bias contributions from optical scattering. The results shown in Figure 4 come directly to mind.

Reply: We would like to thank the reviewer for bringing this important issue to our attention. We agree with the reviewer that in the transmission dark-field configuration, as implemented in our work, the O-PTIR signal intensity can exhibit bias towards particles that scatter incident light more efficiently. This inherent feature could potentially complicate quantitative spectral analysis. In Supplementary Fig. S5, we have now included the DC image of cancer cells for assessing potential bias attributed to optical scattering. Additionally, we also collected DC images in the measurement of mouse brain tissues (Supplementary Fig. S8).

In Supplementary Fig. S5, two focal planes within a cell are shown. Panel (a) focuses on the cell nucleus layer, primarily composed of proteins. In this region, the demodulated AC signal exhibits minimal deviation from the DC channel, indicating relatively low optical scattering effects. In contrast, panel (b) looks into the cell's bottom region, which contains a large number of lipid droplets. Here, optical scattering becomes more pronounced. The strong OPT signal manifests as “holes” in the DC channel due to the strong light scattering by lipid droplets.

The optical scattering effect cannot be readily normalized by simply dividing the AC signal by the DC signal. To address this issue, algorithms such as multiplicative scatter correction (MSC) and standard normal variate (SNV) are typically utilized to correct spectral distortions arising from multiplicative scattering, which is commonly encountered when samples contain particles of varying sizes. An ideal and accurate way to determine modulation depth at each pixel without bias would be to leverage the digitization of transient thermal traces and calculate the depth of modulation directly from the traces at each pixel.

In the revised manuscript, we added a discussion of optical scattering and its influence on complicating quantitative spectral analysis in the cancer cell subsection:

“It's important to recognize that the detected OPT signal intensity is influenced by both the scattered light intensity and the modulation depth resulting from photothermal absorption. This characteristic indicates the potential for bias in the determination of absolute chemical concentrations due to optical scattering. In Supplementary Fig. S5, we present transmission

images (DC signals) without SWIR beam excitation, alongside OPT images (AC signals) of single cancer cells. This comparison allows for a comprehensive assessment of the potential bias. Consequently, the subsequent chemical decomposition via OPT microscopy is carried out in relative concentration (semi-quantitative) measurements, rather than precise absolute quantitative values.”

6. While the authors are certainly correct in concluding that overtone and combination bands exhibit reduced attenuation of the pump and probe beams from lower absorption cross-sections, that reduced cross-section affects not only the background but the signal as well. Reducing both the signal and the noise proportionally does not improve the signal-to-noise ratio. The manuscript should be updated to include a more thorough discussion of the conditions for which major SNR enhancements are anticipated to arise (if at all) relative to O-PTIR with mid-infrared excitation. The results shown in Figure 3 suggest relatively low SNR and spatial resolution compared to previously reported O-PTIR measurements.

Reply: We would like to thank the reviewer for this comment. The much-reduced water absorption in the SWIR window opens possibilities for imaging deep compared to O-PTIR when measuring biological samples. We agree with the reviewer’s observation that the signal-to-noise ratio will not significantly improve with OPT, assuming employing the same probe laser source (similar noise levels from lasers). Figure 3 demonstrates OPT imaging of polymer nanostructures in an air environment to emphasize the versatility of OPT across various conditions. However, we acknowledge that O-PTIR excels specifically in air conditions, and we do not intend to claim that OPT can outperform O-PTIR in such settings. Both techniques have their unique advantages and are ideally suited for specific experimental conditions and sample types.

7. The spectroscopic window accessible by NIR is dominated by overtones and combination bands incorporating H-stretching motions. This window historically lacks high selectivity, consistent with the highly overlapping peak shapes shown in Figure 4C. The authors are encouraged to include a greater discussion of the limitations associated with decomposing signals from overlapping spectral features, particularly if image contrast scales not just with absorption alone but also with local optical scattering. Specifically, what level of SNR is necessary and over what spectral range for confidently assigning composition on a per-pixel or per-segment basis?

Reply: We appreciate the reviewer for the insightful comment that helped us improve the manuscript. Indeed, biomolecules in the overtone region often have overlapping spectral peaks, which mathematically results in uncertainties in linear spectral decomposition using the standard least squares fitting inversion. To address this problem, we introduced a model-based regularization, namely the local chemical sparsity (i.e., pixel-wise LASSO), to the least squares fitting inversion. The regularization is based on the physical observation that under the diffraction limit of ~ 400 nm, for a highly heterogeneous biological sample, only a few species often make dominant contributions at each pixel. This method has recently been used to perform high-content unmixing of hyperspectral stimulated Raman scattering (SRS) images in the crowded C-H region (Tan, Y., Lin, H., & Cheng, J. X. *Science Advances*, 2023, 9(33), eadg6061), which has shown robustness under spectral overlapping conditions like the overtone window. Bias induced by local

optical scattering indeed remains a challenge for photothermal lensing detection, and it cannot be addressed by pixel-wise LASSO. One possible future direction is to leverage the water channel (assuming the water to be homogeneous in the sample) as a calibration metric to compensate for the scattering bias in other chemical channels.

Heuristically, we can determine the necessary SNR by analyzing the image quality of the spectral unmixing output maps. Under sufficient SNR, they will contain reasonable morphological features standing out of the background. Otherwise, the output maps will contain no morphologies regardless of the sparsity level. As for selecting the necessary spectral coverage for robust spectral unmixing, we covered the entire second overtone window to ensure all the spectral features were collected. Since the number of separable components is much less than the spectral frames, we can perform feature selection in the future to sample only the significant spectral features to improve the efficiency.

Reviewer #2 (Remarks to the Author):

The authors presented an interesting approach that suggests minimizing the water signal using SWIR in a photothermal approach. They argued that previously published approaches utilizing visible and FTIR methods have significant limitations. The novelty lies in combining the photothermal approach with hyperspectral SWIR. However, the manuscript fails to acknowledge that hyperspectral SWIR has been used for almost a decade to minimize water and scattering in deep tissue, which is a major limitation. It remains unclear whether adding photothermal to the currently used hyperspectral SWIR provides significant value.

Overall, the manuscript is technically solid, and the idea of combining photothermal with SWIR is novel, potentially acceptable if the authors demonstrate the advantages directly. That is the major issue.

Reply: We appreciate the comprehensive evaluation of our manuscript by the reviewer. We agree with the reviewer's emphasis on the extensive use of hyperspectral SWIR imaging techniques, including hyperspectral reflectance or transmission imaging, in predictive and classificatory analysis of chemical composition. We have added relevant references to the Introduction section.

Meanwhile, we would like to point out that direct SWIR spectroscopy, either in transmission or reflection mode, cannot pick up small objects on the nanoscale, where the tiny absorption ($< 1\%$) is buried in the noise of transmitted or reflected photons.

Photothermal detection, as a pump-probe approach, elegantly addresses this issue. In our method, by comparing the scattered probe photons between the SWIR-on and SWIR-off state, a strong signal from subjects as small as 200 nm can be seen with a clean contrast (Figure 2). In parallel, the use of a visible probe at 520 nm also provides much improved spatial resolution. These advantages expand the capability of SWIR spectroscopy. For example, our OPT method is able to visualize subcellular features. Our method is also applicable to materials such as nano-plastics or nanoscale organic contaminants in semiconductors, and small drug particles in pharmaceutical products.

In summary, our research advances hyperspectral SWIR techniques in the aspect that the integration of a photothermal detection scheme significantly enhances spatial resolution and detection sensitivity of SWIR spectroscopic imaging. As outlined in the introduction, photothermal microscopy serves as a highly sensitive pump-probe technique capable of detecting minute thermal gradients induced by chemical-specific absorption. Consequently, the probe beam's intensity modulation directly reflects photothermal absorption and carries precise local chemical information without any signal loss. Moreover, employing a high-numerical-aperture objective to collinearly and tightly focus both beams ensures superior spatial resolution and, concurrently, establishes a high local field density, thereby augmenting the detection sensitivity.

Several minor technical issues also need to be addressed:

1. The authors should mention that the SWIR region is not uniform, and large penetration depth is only possible at certain wavelengths. They can refer to existing literature that measures depth versus each wavelength. However, for cell samples, this might not be an issue.

Reply: Thanks to the reviewer for this helpful suggestion. We have now incorporated a specific mention of this wavelength dependency in the Introduction section and added two references that studied penetration depth vs. SWIR wavelength. The typical variation range of penetration depth, as elucidated in these references, spans several millimeters. Consequently, the 100- μm imaging depth emphasized in our study remains robust and is not anticipated to be significantly affected.

2. Please explain why the LASSO method was chosen for unmixing and its advantages over other techniques.

Reply: We thank the reviewer for bringing up this point, which resonates with Comment #7 from Reviewer 1. As mentioned in the previous question, overtone signals are spectrally broad, which makes it unstable to mathematically decompose the raw hyperspectral image into chemical maps. To address this challenge, we sought to use regularized inversion, specifically the pixel-wise LASSO, to improve the robustness of the conventional least squares fitting. The regularization is based on a physical observation that for a highly heterogeneous bio-sample, under the diffraction limit of 400 nm, only a few components are making dominant contributions. This method has recently been used to perform high-content unmixing of hyperspectral stimulated Raman scattering (SRS) images in the crowded C-H region (*Tan, Y., Lin, H., & Cheng, J. X. Science Advances, 2023, 9(33), eadg6061*), which has shown robustness under spectral overlapping conditions like the overtone window.

3. Washing cells with D₂O to minimize water absorption might lead to cell death, precluding live cell imaging. Please comment on this concern, as the same might apply to *C. elegans*.

Reply: We appreciate the reviewer for bringing up this important concern. We agree that an excessive amount of D₂O can induce toxicity and potentially result in cell death. Thus, this D₂O-based strategy is suitable for fixed-cell measurements. In the case of bacterial samples, the threshold for D₂O usage is approximately 70%, while for mammalian cells, a safe D₂O threshold typically ranges from 20% to 25%. When investigating dynamics within living cells, it remains essential to utilize a regular phosphate-buffered saline (PBS) as a medium. Hyperspectral analysis is then performed to effectively remove the water background.

4. The methods section should include more technical details, such as the source of albumin, triglycerides, and specifics of the NIR/VIS/SWIR spectrophotometer.

Reply: Thanks to the reviewer for pointing this out. In the revised manuscript, we have added the source of bovine serum albumin and triglycerides in the Methods section. We also included a short paragraph outlining the parameters involved in utilizing a commercial UV-Vis-NIR spectrophotometer for spectra collection.

“In the demonstrated spectral unmixing of biological specimens in water environment, OPT spectra of bovine serum albumin (BSA, purity $\geq 96\%$, Sigma-Aldrich), triglyceride (TAG, purity $\geq 97\%$, Sigma-Aldrich), and deionized water were collected, serving as the pure spectral references for protein, fatty acids, and water, respectively.”

“**Collection of NIR absorption spectra via UV-Vis-NIR spectrophotometer** Standard NIR absorption spectra for both polystyrene and ethylene glycol were acquired utilizing the Cary

5000 UV-Vis-NIR Spectrophotometer (Agilent) following established protocols. Polystyrene ($M_w = 35,000$, Sigma-Aldrich) was dissolved in toluene, and 3 mL of the resulting solution was transferred into a quartz cuvette for insertion into the spectrophotometer. An absorption spectrum of toluene was acquired and subsequently utilized for baseline correction. Similarly, 3 mL ethylene glycol (purity 99.8%, Sigma-Aldrich) was introduced into a separate quartz cuvette, with ambient air as the reference baseline. The measurement parameters include a spectral bandwidth (SBW) of 2 nm, an integration time of 0.1 s, and a wavelength interval of 1 nm.”

5. In Figure 2, "spectral fidelity" usually refers to spectral calibration. Please check if this term is correct in the given context.

Reply: We would like to thank the reviewer for suggesting this point. As per our understanding, “spectral fidelity” describes the extent to which OPT spectra align with established standard or ground truth NIR spectra. It also refers to how closely OPT results match those obtained through conventional approaches. We have carefully examined the usage of "spectral fidelity" in relevant literature and have found it to be an accepted term describing high spectral accuracy. For example, in the paper published by *Freudiger et al. Nature Photonics 8, 153–159 (2014)*, the authors described stimulated Raman scattering as having a better **spectral fidelity** compared with coherent anti-Stokes Raman scattering when validating against spontaneous Raman.

6. The shape of the spectra shown by OPT differs somewhat from those previously published by Cao et al. *Journal of Biomedical Optics* 18.10 (2013): 101318.

Reply: We appreciate the reviewer for bringing up this observation. Upon a careful comparison with the NIR spectra published in the work by Cao et al., we have identified two differences.

First, we observed a dissimilarity in the overall baseline shape of the NIR spectra. This discrepancy is primarily attributed to the power drop experienced in the long wavelength window of our laser, in conjunction with the lower transmission efficiency of the glass rods used for pulse chirping in this window. Notably, the spectra presented in Fig. 4c (BSA, TAG, and water) were not power-normalized, as both the reference spectra and cell images were acquired sequentially under identical power conditions. In contrast, the water spectrum (power-normalized) shown in Fig. S4 closely aligns with the spectral shape reported in Cao’s paper. The typical laser power profile in Fig. S4 was shown as a reference.

The second difference we observed is the presence of small, narrow peaks in the spectra by Cao et al. This arises from the spectral resolution of our femtosecond ultrafast laser. The estimated spectral width of the chirped picosecond pulse is approximately 10 nm, resulting in a somewhat smoothed shoulder peak. For instance, in Fig. 3c of Cao’s work, corn oil exhibits a minute shoulder peak at ~ 1155 nm, corresponding to the 2nd overtone of unsaturated C-H bonds. In our OPT spectra depicted in Fig. 4c, a similar peak centered at ~ 1160 nm appears broader and smoother. This characteristic is inherent to ultrafast laser pulses. We anticipate that future investigations into OPT microscopy using a supercontinuum laser or nanosecond laser will circumvent this limitation and yield a higher spectral resolution.

Thanks to the reviewer for sharing this work, and we have now cited this paper by Cao et al. in our manuscript as a benchmark reference.

Reviewer #3 (Remarks to the Author):

This paper presents a version of photothermal (PT) microscopy that uses overtone vibrations as the primary source of contrast. This work is a PT implementation of a similar idea pursued earlier by the same group, which was based on overtone excitation with photoacoustic (PA) detection. The PT technique itself, based on the excitation of fundamental molecular vibrations (not the overtones), has been a focus of the group for the last 8 years or so. Here, the authors claim that overtone PT (OPT) represents several advantages compared to their mid-IR PT work: 1) reduced background from water and 2) higher resolution because better (refractive) lenses can be used.

The paper is interesting, and the data shows the technique has merit. In a sense, it is a new variation of the PT family of techniques. Although certainly new, some of the claims in this work come across as rather strong. In addition, the following points are noted:

We would like to thank the reviewer for taking the time to evaluate our manuscript and for the constructive comments. Please see below for the actions we have taken to address the comments and suggestions.

1) While water absorption in the mid-IR range is substantial, it is also present in the NIR. In addition, the fundamental absorptions in the mid-IR are strong, whereas the 2nd overtones in the NIR are comparatively weak. This means that the weaker (and broadened) NIR features appear on a water background, which also reduces contrast. Therefore, it is not sufficient to state that OPT has an automatic advantage over mid-IR without some sort of quantification in terms of contrast etc.

Reply: We appreciate the insightful comment provided by the reviewer. A major advantage of OPT over mid-IR photothermal microscopy is that OPT can penetrate deeper into an aqueous environment owing to the much smaller water absorption in the SWIR window.

To obtain water absorption coefficients in SWIR and mid-IR windows, we referred to the optical constants of water published in *Hale, G. M., & Querry, M. R. Applied Optics, 1973, 12(3), 555-563*. We chose 1.2 μm and 6.0 μm , which correspond to the second overtone of the C-H bond and the most intense stretching vibrations of the Amide I band in mid-IR window, respectively. The water extinction coefficient (κ) was found to be $\kappa_{1.2 \mu\text{m}} = 9.89 \times 10^{-6}$ and $\kappa_{6.0 \mu\text{m}} = 0.107$. By calculating the absorption coefficient (α) using the formula $\alpha = 4\pi\kappa/\lambda$, the water absorption difference between 6.0 μm and 1.2 μm is $\alpha_{6.0 \mu\text{m}}/\alpha_{1.2 \mu\text{m}} = 2164$. Thus, because of the much lower water absorption, OPT microscopy can see deeper into biological tissues than mid-IR microscopy, as we demonstrated using a mouse brain slice in newly added Fig. 7.

2) The images in Figure 4 reveal a substantial water background. In addition, the "protein" image appears to be affected by transmission effects, which goes beyond the more quantitative chemical contrast. These issues are not discussed clearly in the manuscript.

Reply: We would like to thank the reviewer for providing this valuable comment. The water background in cell measurements is within expectation and can be removed through appropriate data post-processing. The cancer cell images in Figure 4 demonstrate the capability of OPT

microscopy to effectively unveil intracellular metabolites when a water background is present, with the aid of spectral unmixing analysis.

Additionally, we are fully aware that the OPT signal intensity is influenced by both the scattered light intensity (transmission image) and the modulation depth resulting from photothermal absorption. This dual dependency implies the potential for bias in determining absolute chemical concentrations, primarily due to the effects of optical scattering. In the revised manuscript, we included the transmission image (DC signal) and the OPT images (AC signal) of cancer cells for reference, illustrating the potential bias caused by optical scattering (Supplementary Fig. S5). We have addressed this effect in the discussion of Figure 4 and revised our statement regarding OPT being a quantitative approach to provide a more accurate representation of its capabilities and limitations. The added discussion reads as follows:

“It's important to recognize that the detected OPT signal intensity is influenced by both the scattered light intensity and the modulation depth resulting from photothermal absorption. This characteristic indicates the potential for bias in the determination of absolute chemical concentrations due to optical scattering. In Supplementary Fig. S5, we present transmission images (DC signals) without SWIR beam excitation, alongside OPT images (AC signals) of single cancer cells. This comparison allows for a comprehensive assessment of the potential bias. Consequently, the subsequent chemical decomposition via OPT microscopy is carried out in relative concentration (semi-quantitative) measurements, rather than precise absolute quantitative values.”

3) "These findings demonstrate the promising potential of OPT microscopy to provide unambiguous and quantitative chemical information in large and complex multicellular biological systems, irrespective of the imaging media used." Too strong of a statement, as the medium here is actually D₂O, which was artificially introduced to suppress the background from water.

Reply: Thanks to the reviewer for pointing out this issue. We have toned down this concluding statement. In the revised manuscript, this sentence is modified as:

“These findings demonstrate the promising potential of OPT microscopy to provide quantitative insight into chemical information in large and complex multicellular biological systems, with a flexibility of choosing H₂O PBS or D₂O PBS as the media.”

4) The *C. elegans* demonstration is nice, but this specimen is very small and thin, which means that the reader still cannot determine the OPT advantage over mid-IR imaging in aqueous samples. In the mid-IR, samples can still be imaged at depths up to ~20 μm, so the current demonstration does not necessarily demonstrate an advantage in terms of deeper imaging in tissues.

Reply: We appreciate the reviewer for the insightful comment that helped us improve the manuscript. In the revised manuscript, we have conducted depth-resolved OPT imaging on a highly scattering medium, i.e. mouse brain slices with a thickness of 200 μm, to demonstrate the penetration depth achievable through OPT microscopy. The results are presented in Fig. 7.

Meanwhile, for comparative analysis, another adjacent brain slice was prepared under identical parameters and retained for mid-IR photothermal measurement. These images are included in Supplementary Fig. 7. We intentionally avoided the strongest water absorption peak at ~ 1650 cm^{-1} and selected the Amide II band at 1553 cm^{-1} for imaging in the mid-IR window. Both the OPT and mid-IR tests employed the same co-propagation and forward detection configuration to ensure a fair and accurate comparison.

The obtained results suggest that mid-IR reaches a penetration depth of ~ 40 μm , while OPT can go to approximately 100 μm . This verifies that water absorption substantially reduces penetration depth and OPT microscopy outperforms mid-IR photothermal microscopy in terms of imaging deeper into biological tissues.

REVIEWER COMMENTS

Reviewer #1 (Remarks to the Author):

Summary

The authors have made substantial changes to the manuscript to address the reviewers' comments. Notably, the authors revised the Introduction to more properly place the work in context, and added a figure (Figure 7) to highlight the depth-penetration capabilities of the method. These changes did not fully address concerns raised relative to the use of NIR transmission / diffuse reflectance hyperspectral imaging, the sensitivity relative to O-PTIR, or the specificity of overtone spectroscopy. However, the manuscript is well-written and structured, and the newly added brain tissue results help to strengthen the authors' point. The technique itself utilizing a easily accessible tunable NIR laser source with common-path optics in a beam-scanning configuration is a substantial and important advantage over the most commonplace sample-scanning approaches for O-PTIR. The manuscript is recommended for publication following efforts to address the following points.

Major comments

1. In the response, the authors acknowledge that claims of improved sensitivity states in the original manuscripts were not justified, with the core advantages instead around penetration depth. However, the Introduction and Conclusions still refer numerous times to "superior sensitivity" and other superlatives that are misleading given the acknowledgement by the authors of a negligible SNR improvement. The authors are encouraged to review the manuscript again to remove comments suggesting improvements in sensitivity by OPT.
2. The authors updated the manuscript to include comparisons between O-PTIR and OPT with the additional brain images, demonstrating a $\sim 2x$ improvement in penetration depth ($\sim 40 \mu\text{m}$ by O-PTIR vs. $\sim 80 \mu\text{m}$ by OPT). These results support the improvement in penetration depth by OPT relative to O-PTIR but raise two additional questions: i) why is the improvement only $\sim 2x$ rather than the 20-5000x expected based on the IR cross-section?, and ii) how do these results compare to simple measurements of the transmitted or diffuse-reflected IR? It is curious that the latter results were not included in the data set, as they should in principle be accessible simply by addition of a photodiode detector for parallel measurements. As such, the question of comparison with NIR spectroscopic imaging remains open.
3. The authors included statements in the Introduction claiming that NIR imaging is typically limited in resolution to 10s of micrometers, which is not accurate. Numerous studies from Bhargava et al. have demonstrated much better spatial resolution than this without even needing to extend outside of the mid-IR band. Resolution on the order of half the NIR wavelength ($\sim 1.1 \mu\text{m}/2 = \sim 550 \text{ nm}$) should be expected in the beam-scanning measurements described in this work with an NA = 1.1 60x water immersion objective.
4. The authors did not fully address the comment #7 about the chemical specificity of the NIR region, and the general challenges of extracting selective chemical information from spectral decomposition of highly overlapping broad spectral features. It does not appear that changes to the manuscript were made to address this point. Given that the SNR is not expected or observed to improve in OPT relative to O-PTIR, the net effect is a loss in classification accuracy in OPT by nature of the higher required SNR in the presence of broad, overlapping spectral bands. The authors are encouraged to revise the manuscript to fairly evaluate the limitations of the method in light of the overlapping spectral features common in NIR spectroscopy.

Reviewer #2 (Remarks to the Author):

The authors responded well to all comments. The manuscript can now be accepted for publication.

Reviewer #3 (Remarks to the Author):

I have read the revised manuscript and the response by the authors. I have no further comments.

Reviewer #1 (Remarks to the Author):

The authors have made substantial changes to the manuscript to address the reviewers' comments. Notably, the authors revised the Introduction to more properly place the work in context and added a figure (Figure 7) to highlight the depth-penetration capabilities of the method. These changes did not fully address concerns raised relative to the use of NIR transmission / diffuse reflectance hyperspectral imaging, the sensitivity relative to O-PTIR, or the specificity of overtone spectroscopy. However, the manuscript is well-written and structured, and the newly added brain tissue results help to strengthen the authors' point. The technique itself utilizing an easily accessible tunable NIR laser source with common-path optics in a beam-scanning configuration is a substantial and important advantage over the most commonplace sample-scanning approaches for O-PTIR. The manuscript is recommended for publication following efforts to address the following points.

We deeply appreciate the reviewer for the comprehensive review of our revised manuscript and the constructive suggestions that further strengthen the manuscript. Please see below for the actions we have taken to address the comments.

1. In the response, the authors acknowledge that claims of improved sensitivity states in the original manuscripts were not justified, with the core advantages instead around penetration depth. However, the Introduction and Conclusions still refer numerous times to “superior sensitivity” and other superlatives that are misleading given the acknowledgement by the authors of a negligible SNR improvement. The authors are encouraged to review the manuscript again to remove comments suggesting improvements in sensitivity by OPT.

Reply: We are grateful to the reviewer for this suggestion and would like to respectfully provide some clarification regarding our claim of improved sensitivity. Our claim is based on a comparison with existing shortwave infrared (SWIR) imaging techniques. Our intention does not involve surpassing O-PTIR in terms of detection sensitivity. Given that both OPT and O-PTIR are based on photothermal lensing detection and considering the approximately two-to-three orders of magnitude cross-section difference between the second overtone and the fundamental transition, O-PTIR will offer superior sensitivity for thin samples. We recognize that drawing simultaneous comparisons between (1) OPT and conventional SWIR hyperspectral imaging and (2) OPT and O-PTIR in our manuscript might create ambiguity.

The significance of our work lies in improving the sensitivity and resolution of the current SWIR spectroscopic imaging methods via photothermal pump-probe detection. Direct SWIR spectroscopy, either in transmission or reflection mode, cannot pick up small objects on the sub-micron scale. In such scenarios, the tiny absorption (< 1%) is often buried in the noise of transmitted or reflected photons. Conversely, as demonstrated in Figure 2 of the manuscript, OPT provides strong signals from subjects as small as 200 nm with a clean contrast. This advancement broadens the applicability of SWIR spectroscopy, enabling the visualization of subcellular features, characterization of materials such as nano-plastics or nanoscale organic contaminants in semiconductors, and identification of small drug particles in pharmaceutical products.

To address potential ambiguity, we have moved the discussion of visible and mid-IR photothermal microscopy from the Introduction section to the Discussion section. Within the Discussion section, we have revised the first part and also added a table (Supplementary Table S1) comparing the resolution and penetration depth among various SWIR hyperspectral imaging methods, O-PTIR, and OPT microscopy. The comparison indicates that OPT creates new opportunities in two aspects: (1) it supplements SWIR spectroscopic imaging techniques by providing improved resolution and sensitivity; (2) meanwhile, functioning as a photothermal method, it bridges the gap between visible photothermal microscopy and mid-infrared photothermal microscopy (O-PTIR), affording the capability to discern chemical specifics of biological tissues with enhanced penetration. The revised Discussion is included below:

“Shortwave infrared (SWIR) imaging has great potential for the visualization of biological structures and metabolites deep inside opaque tissues. A panel of SWIR spectroscopy and imaging technologies, such as hyperspectral transmittance and reflectance⁸, photoacoustic¹⁴, and OCT (optical coherence tomography)-based imaging⁴², have been widely adopted for their crucial roles in biomedical functional and chemical analysis. To accommodate a larger field-of-view and deeper penetration, current SWIR imaging approaches often sacrifice lateral resolution and detection sensitivity. In this work, we introduced overtone photothermal (OPT) microscopy, utilizing SWIR excitation of overtone bands coupled with a visible beam to probe thermal effects. OPT microscopy achieves both high resolution and high sensitivity, benefiting from its pump-probe detection scheme, the utilization of a visible probe, and a high numerical aperture (NA) objective. As demonstrated in the results of polymer nanoparticles, OPT microscopy offers a lateral resolution of 405 nm and can clearly detect individual 200-nm PMMA beads. This is the first time to achieve simultaneous high spatial resolution and high sensitivity imaging within the SWIR region. We should acknowledge that such advantages come with a compromised imaging depth of ~ 100 μm . Nevertheless, the enhanced imaging resolution and sensitivity broadens the applicability of SWIR spectroscopy, enabling the visualization of subcellular features, characterization of materials such as nano-plastics or nanoscale organic contaminants in semiconductors, and identification of small drug particles in pharmaceutical products.

Photothermal microscopy is a pump-probe technique that optically detects local thermal gradients with high sensitivity. In photothermal microscopy, molecules or nanoparticles in a sample selectively absorb the pump excitation beam and undergo local temperature increase. The temperature change induces a local variation of the refractive index of the medium, thus altering the transmission or scattering of the probe beam. To date, two main categories of photothermal microscopy have been developed based on the choice of excitation frequency in the visible or mid-infrared. Visible photothermal microscopy leverages the field enhancement of gold nanostructures and targets single light-absorbing nanoparticles, such as single nonfluorescent dye molecules⁴³ and semiconductor nanocrystals⁴⁴. To enable bond-selective photothermal imaging, mid-infrared photothermal (MIP) microscopy offering abundant chemical information in the fingerprint region has been developed.^{21,26,45,46} MIP has been successfully applied to material and life science. Examples include mapping of Fabry–Pérot modes of single metal nanowires⁴⁷ and local cation heterogeneities of perovskites,⁴⁸ probing bacterial metabolic responses at a single-cell level,⁴⁹⁻⁵¹ structural mapping of proteins in cells and tissues^{52,53}, and mapping of enzymatic

activities through mid-infrared photothermal reporters in the silent window⁵⁴. Despite these achievements, MIP microscopy faces a strong absorption of water, resulting in substantial optical losses in thick tissues and a considerable background for measurements in aqueous environments.

In comparison, OPT microscopy provides chemical specificity while circumventing the substantial optical attenuation caused by water absorption in the mid-IR range. This enables versatile imaging applications, from observing living cells in their native environment to achieving deeper penetration into biological tissues. Including hyperspectral unmixing analysis further enhances the chemical resolving power in the SWIR window, where multiple functional groups exhibit overlapping absorption peaks. Supplementary Table 1 summarizes the attributes of current SWIR imaging methods, MIP microscopy, and OPT microscopy.”

2. The authors updated the manuscript to include comparisons between O-PTIR and OPT with the additional brain images, demonstrating a ~2x improvement in penetration depth (~40 μm by O-PTIR vs. ~80 μm by OPT). These results support the improvement in penetration depth by OPT relative to O-PTIR but raise two additional questions: i) why is the improvement only ~2x rather than the 20-5000x expected based on the IR cross-section? and ii) how do these results compare to simple measurements of the transmitted or diffuse-reflected IR? It is curious that the latter results were not included in the data set, as they should in principle be accessible simply by the addition of a photodiode detector for parallel measurements. As such, the question of comparison with NIR spectroscopic imaging remains open.

Reply: We appreciate the helpful feedback provided by the reviewer. Addressing the first point, as OPT is essentially a pump-probe process, the penetration depth is determined by both the SWIR beam and the 520 nm probe beam. Extensive studies have revealed that visible beams tend to undergo stronger multiple-scattering and photon diffusion in a turbid medium compared with long wavelengths. This results in a distorted wavefront and a broadened focal spot of the probe beam, thereby practically limiting the OPT signal intensity and, consequently, the effective penetration depth. Supplementary Fig. S8’s transmission images indicate that the lateral resolution degradation occurs beyond a depth of 80 μm .

Regarding the second point, we sincerely thank the reviewer for the suggestion of acquiring transmission images with the SWIR beam. In response, we have obtained depth-resolved transmittance (DC) images of mouse brain slices at 1190 nm and included the results in Supplementary Fig. S9, demonstrating a penetration depth of approximately 285 μm . This depth limitation is attributed to the 0.28 mm working distance of our NA = 1.2 water objective. This data confirms that the penetration depth in our current OPT configuration is not limited by the SWIR beam, as previously noted in Comment (i).

When comparing the SWIR transmission images to OPT images, certain limitations become apparent. The hyperspectral SWIR transmittance data fails to provide reliable chemical information even after power normalization, due to a considerable DC transmission background with buried < 1% attenuation. Additionally, the SWIR transmission images exhibit poorer lateral resolution and axial sectioning capability. As a result, the pump-probe photothermal approach is

a more effective method for enhancing chemical sensitivity and axial resolution in contrast to conventional hyperspectral transmittance imaging.

We have incorporated the following discussion on penetration depth into brain tissue imaging within the Results section:

“It's worth noting that, in the current configuration of OPT microscopy, the penetration depth is primarily determined by the visible probe beam rather than the SWIR beam. The 520 nm beam encounters more pronounced wavefront distortion and focus broadening than the SWIR beam while transmitting through a turbid medium, due to considerable multiple scattering effects. As demonstrated in Supplementary Fig. S9, the SWIR beam exhibits the capability to penetrate through the entire working distance (0.28 mm) of our water objective without substantial beam focus distortion.”

3. The authors included statements in the Introduction claiming that NIR imaging is typically limited in resolution to 10s of micrometers, which is not accurate. Numerous studies from Bhargava et al. have demonstrated much better spatial resolution than this without even needing to extend outside of the mid-IR band. Resolution on the order of half the NIR wavelength ($\sim 1.1 \mu\text{m}/2 = \sim 550 \text{ nm}$) should be expected in the beam-scanning measurements described in this work with an NA = 1.2 60X water immersion objective.

Reply: We are grateful for this valuable input from the reviewer. We apologize for any confusion arising from our prior Introduction regarding the mention of “scanning mode”. To clarify, our intended reference was to a wavelength-scanning mode specifically utilized for registering hyperspectral image stacks, rather than beam scanning. Conventional NIR hyperspectral imaging approaches often prioritize a large field-of-view and deep penetration, typically employing wide-field or line-scan schemes with a low NA focusing element and camera detection. However, these methods commonly sacrifice diffraction-limited resolution.

We have now revised the description of SWIR hyperspectral reflectance/transmittance imaging approaches for better accuracy and clarity. Additionally, we have referenced the work by Bhargava et al. as noteworthy examples of diffraction-limited spectroscopic imaging studies in the Introduction.

The revised context now reads as follows in the first paragraph of the Introduction:

“SWIR hyperspectral imaging, which measures reflectance or transmittance, primarily serves in the qualitative spectral characterization of samples at the macroscale, focusing on properties related to absorption and scattering. Employing wide-field or line-scan configurations with low-NA focusing elements and camera-based detection, this technique often sacrifices spatial resolution in favor of a large field-of-view (FOV) due to the limited space-bandwidth product. To enlarge the FOV with a limited number of camera pixels, the camera pitch size was not optimized to the Nyquist Limit, therefore leading to resolutions on the order of ten micrometers. Conversely, diffraction-limited spectroscopic imaging can be attained through the implementation of a laser point-scanning design.^{15,16”}

4. The authors did not fully address the comment #7 about the chemical specificity of the NIR region, and the general challenges of extracting selective chemical information from spectral decomposition of highly overlapping broad spectral features. It does not appear that changes to the manuscript were made to address this point. Given that the SNR is not expected or observed to improve in OPT relative to O-PTIR, the net effect is a loss in classification accuracy in OPT by nature of the higher required SNR in the presence of broad, overlapping spectral bands. The authors are encouraged to revise the manuscript to fairly evaluate the limitations of the method in light of the overlapping spectral features common in NIR spectroscopy.

Reply: We would like to thank the reviewer for providing this insightful comment. We apologize for our oversight in not addressing the related discussions in the previous revision. In response, we've now added substantial discussions to evaluate the rationale behind using the LASSO algorithm for spectral unmixing and the potential challenges that may impact the method's quantitative capacity. Specifically, we talked about challenges arising from optical scattering, the overlapping nature of spectral features, and the consideration in selecting spectral frames. The added text is included below for your reference (5th paragraph in Discussion):

“One limitation of the SWIR spectroscopic window is spectral peak overlapping, which results in crosstalks between chemical species in hyperspectral image unmixing, impairing the specificity in biological systems. In this work, we sought to improve the specificity through synergistic innovations in instrumentation and data science. Compared with traditional SWIR technologies, OPT can perform hyperspectral imaging at a subcellular resolution of 405 nm. We reason that under such resolutions, for a spatially heterogeneous biological system, only a few species have dominant contributions within each laser scanning spot. This physical prior knowledge, which we refer to as "local chemical sparsity", is mathematically translated as an L1-norm regularizer on the concentration vector at each pixel (i.e., pixel-wise LASSO), thereby suppressing signal crosstalks effectively. Nevertheless, several challenges remain in advancing the quantitative multiplexing capability of OPT spectroscopic imaging. First, photothermal lensing detection is known to scale with both absorption and local optical scattering, the latter of which can result in biased concentrations after unmixing. This issue could be potentially addressed by either leveraging the water unmixing channel as a local calibration metric to compensate for scattering bias, or through quantitative phase detection, which scales solely with refractive index. Second, the broad spectral peaks in the SWIR region still pose high demand for the SNR of the data. Heuristically, the necessary SNR can be determined by inspecting the quality of the output concentration maps. With sufficient SNR, maps of different chemical species should reflect the distinct morphological structures of subcellular organelles that differ from the water background channel, such as lipid droplets in the fatty acid channel. Lastly, to capture all the possible spectral differences amidst broad spectral peaks, we measured the entire second overtone window permitted by the laser tuning range. This approach is not optimized, as the number of separable species is much fewer than the spectral frames and theoretically requires fewer spectral frames. In the future, we aim to perform recursive feature elimination to selectively measure important spectral bands that contribute most to the differentiation of species.”

REVIEWERS' COMMENTS

Reviewer #1 (Remarks to the Author):

The authors have made considerable changes to the manuscript in response to the most recent recommendations upon review. Specifically, the authors made the following modifications.

1. A more substantial discussion was provided on the sensitivity comparisons between O-PTIR, OPT, and SWIR detection for chemical classification. This discussion nicely provides a fair assessment of the merits of OPT within the framework of alternative existing methodologies.
2. The authors made changes to clarify the importance of scattering for the probe beam in dictating the net penetration depth of the measurement approach.
3. The authors have updated the manuscript to include a more complete discussion of the conditions leading to loss of spectral resolution in SWIR imaging, which are primarily attributed to the need for large FoV imaging with a focal plane array detector and the corresponding loss in NA. Their explanation is not as satisfying as their other responses, as the magnification could always be increased to sacrifice FoV for resolution using a focal plane array detector, or using beam-scanning imaging to perform reflectance/transmittance SWIR microscopy. The decision to make such a sacrifice may well be justified for the purposes of the specific study (e.g., to probe a larger area for improved representative sampling) and is not a fundamental limitation of the approach.
4. The authors have expanded on the description of the approach taken for spectral unmixing. A key challenge remains in spectroscopic analysis in the SWIR spectral range, as many of the spectroscopic peaks tend to be broad and spectrally overlapping. Over a given spectral window, the authors point out the need for high SNR for reliable spectral unmixing, in which they adopted a LASSO approach founded around the use of the L1 norm consistent with the assumption of local chemical sparsity within a given focal volume. The changes made to the manuscript provide an improved acknowledgement on the general challenges of performing chemical classification based on SWIR spectral analysis relative to the more discriminatory "fingerprint" spectral windows in the MIR.

Reviewer #1 (Remarks to the Author):

The authors have made considerable changes to the manuscript in response to the most recent recommendations upon review. Specifically, the authors made the following modifications.

We deeply appreciate the reviewer for the comprehensive review of our revised manuscript. Please see below for the actions we have taken to address the comments.

1. A more substantial discussion was provided on the sensitivity comparisons between O-PTIR, OPT, and SWIR detection for chemical classification. This discussion nicely provides a fair assessment of the merits of OPT within the framework of alternative existing methodologies.

Reply: We are grateful for the reviewer's constructive comments and their acknowledgment of our efforts to provide a comprehensive assessment of these technologies.

2. The authors made changes to clarify the importance of scattering for the probe beam in dictating the net penetration depth of the measurement approach.

Reply: We would like to express our gratitude to the reviewer for acknowledging our efforts to elucidate the scattering effect of the probe beam on the penetration depth in overtone photothermal microscopy.

3. The authors have updated the manuscript to include a more complete discussion of the conditions leading to loss of spectral resolution in SWIR imaging, which are primarily attributed to the need for large FoV imaging with a focal plane array detector and the corresponding loss in NA. Their explanation is not as satisfying as their other responses, as the magnification could always be increased to sacrifice FoV for resolution using a focal plane array detector, or using beam-scanning imaging to perform reflectance/transmittance SWIR microscopy. The decision to make such a sacrifice may well be justified for the purposes of the specific study (e.g., to probe a larger area for improved representative sampling) and is not a fundamental limitation of the approach.

Reply: We would like to thank the reviewer for this insightful feedback. We agree with the reviewer regarding the inherent trade-off between spatial resolution and field-of-view (FOV) in existing SWIR imaging techniques. To address this comment, we have revised the relevant discussion in our manuscript. The updated text is now presented as follows (Line 12 – 17 in the first paragraph of Introduction):

“To probe macroscopic sample areas for improved representative sampling, this technique generally sacrifices spatial resolution in favor of a larger field-of-view (FOV) due to the space-bandwidth product limit, thereby leading to resolutions on the order of ten micrometers. Diffraction-limited spectroscopic imaging can be attained through magnifying the FOV onto the camera or implementing a laser point-scanning design.^{15,16}”

This revision clarifies that while there is a usual compromise in spatial resolution for an expanded FOV, the potential for high-resolution imaging is not precluded for existing SWIR imaging methods and can be realized through specific technical adaptations.

4. The authors have expanded on the description of the approach taken for spectral unmixing. A key challenge remains in spectroscopic analysis in the SWIR spectral range, as many of the spectroscopic peaks tend to be broad and spectrally overlapping. Over a given spectral window, the authors point out the need for high SNR for reliable spectral unmixing, in which they adopted a LASSO approach founded around the use of the L1 norm consistent with the assumption of local chemical sparsity within a given focal volume. The changes made to the manuscript provide an improved acknowledgement on the general challenges of performing chemical classification based on SWIR spectral analysis relative to the more discriminatory “fingerprint” spectral windows in the MIR.

Reply: We appreciate the reviewer's recognition of our expanded discussion on the use of the LASSO approach to address broad and spectrally overlapping peaks in the SWIR spectral range.